# The formation and composition of the Mount Everest plume in winter

Edward E. Hindman[1], Scott Lindstrom[2]

[1]Department Earth and Atmospheric Sciences, The City College of New York, New York, 10031, US

[2]Space Sciences and Engineering Center, University of Wisconsin, Madison, Wisconsin, 53706, US

*Correspondence to*: Edward E. Hindman (ehindman@ccny.cuny.edu)

**Abstract.** Mt. Everest's summit pyramid is the highest obstacle on earth to the wintertime jet-stream winds. Downwind, in its wake, a visible plume can form. The meteorology and composition of the plume are unknown. Accordingly, daily from 1 November 2020 through 31 March 2021 (151 days), we observed real-

time images from a geosynchronous meteorological satellite to identify the days plumes formed. The corresponding surface and upper-air meteorological data were collected. The massif was visible on 143 days (95%), plumes formed on 63 days (44%) and lasted an average of 12 hours. We used the upper-air data with a basic meteorological model to show the plumes formed when sufficiently moist air was drawn into the wake. We conclude the plumes were composed initially of either cloud droplets or ice particles depending

on the temperature. The plumes were not composed of resuspended snow. One plume was observed to glaciate downwind. We estimated snowfall from the plumes may be significant.

## 1 Introduction

Mt. Everest's summit is the highest elevation on earth at 8848 m and its summit pyramid (Figs. 1a and 1b) is the largest obstacle to the upper-air winds. With sufficient flow, a turbulent wake forms downwind of the

pyramid and a visible plume can form in the wake as seen in Fig. 2. The meteorology and composition of the plume have been studied, but have not been determined conclusively. This study is a first-step to determine the plume's meteorology and composition. We studied the plume in winter as have all previous investigators. The previous studies, to our knowledge, are as follows.

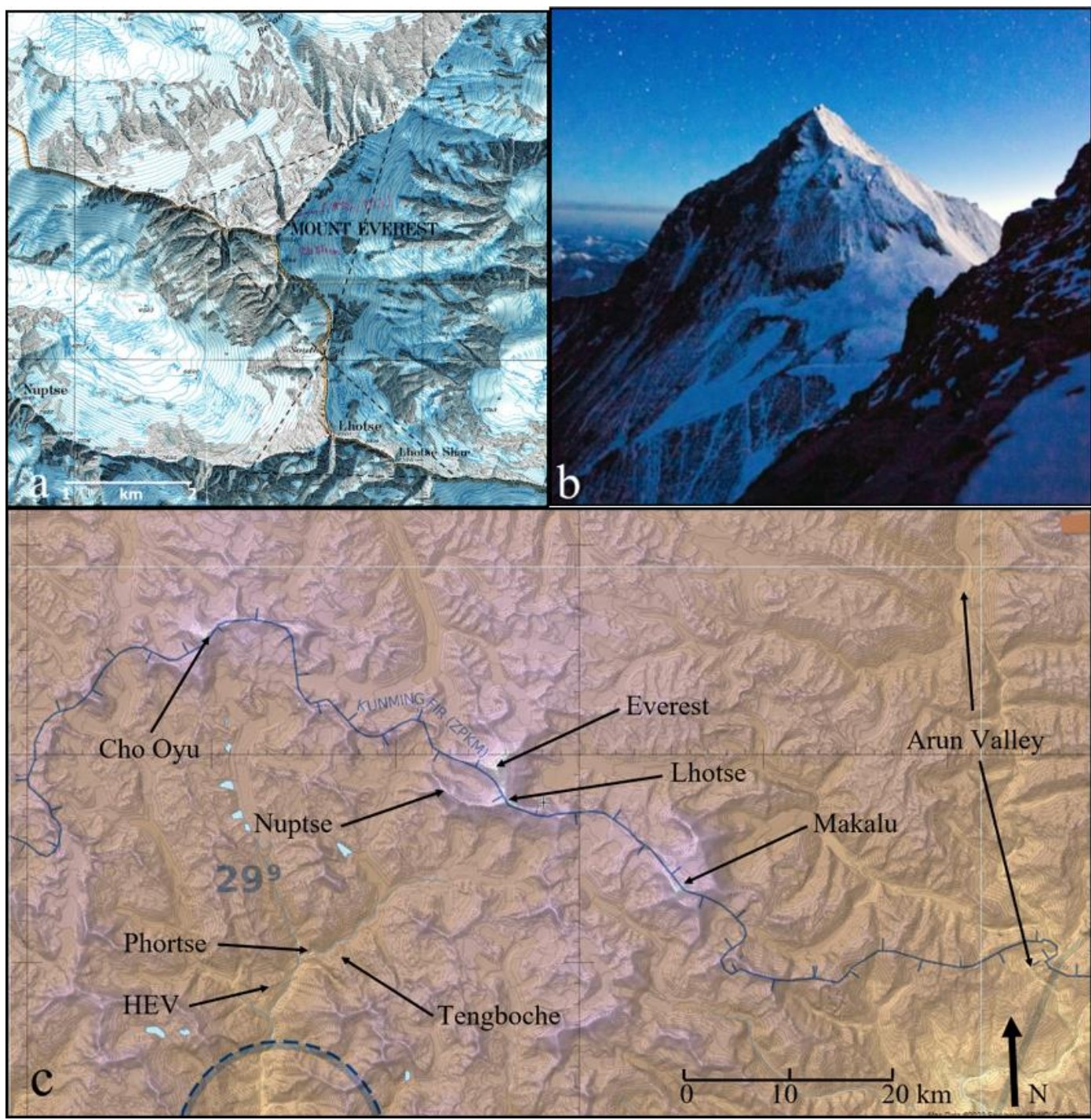

Figure 1. (a) The Mount Everest and Lhotse summit pyramids are outlined. The bases of the pyramids are at an elevation of approximately 7900 m. The summits are, respectively, 8848 m and 8501 m in elevation. The map is from the November 1988 issue of the National Geographic Magazine. (b) The Everest summit pyramid at sunrise in May 2010 as viewed from near the summit of Lhotse (from CoryRichards.com and Anker, et. al. (2013)). (c) The Mount Everest region with the major summits and locations identified (the HEV is the Hotel Everest View; the chart is from skyvector.com).

A January 2004 plume was investigated by Moore (2004) (Fig. 2 - top and middle). He concluded the plume was composed of resuspended snow blown from the peak. He argued that because the atmosphere was too dry the plume could not have been a banner cloud (Douglas, 1928), i.e., a collection of cloud droplets. A

plume photographed by Venables (1989) looks almost identical to Moore's plume (Fig. 2 - bottom). Venables, who was on his way to climb Everest's east face (obscured in the image by the plume), referred to the plume as "the usual plume of cloud and snow, blasted off the summit by the prevailing westerlies".

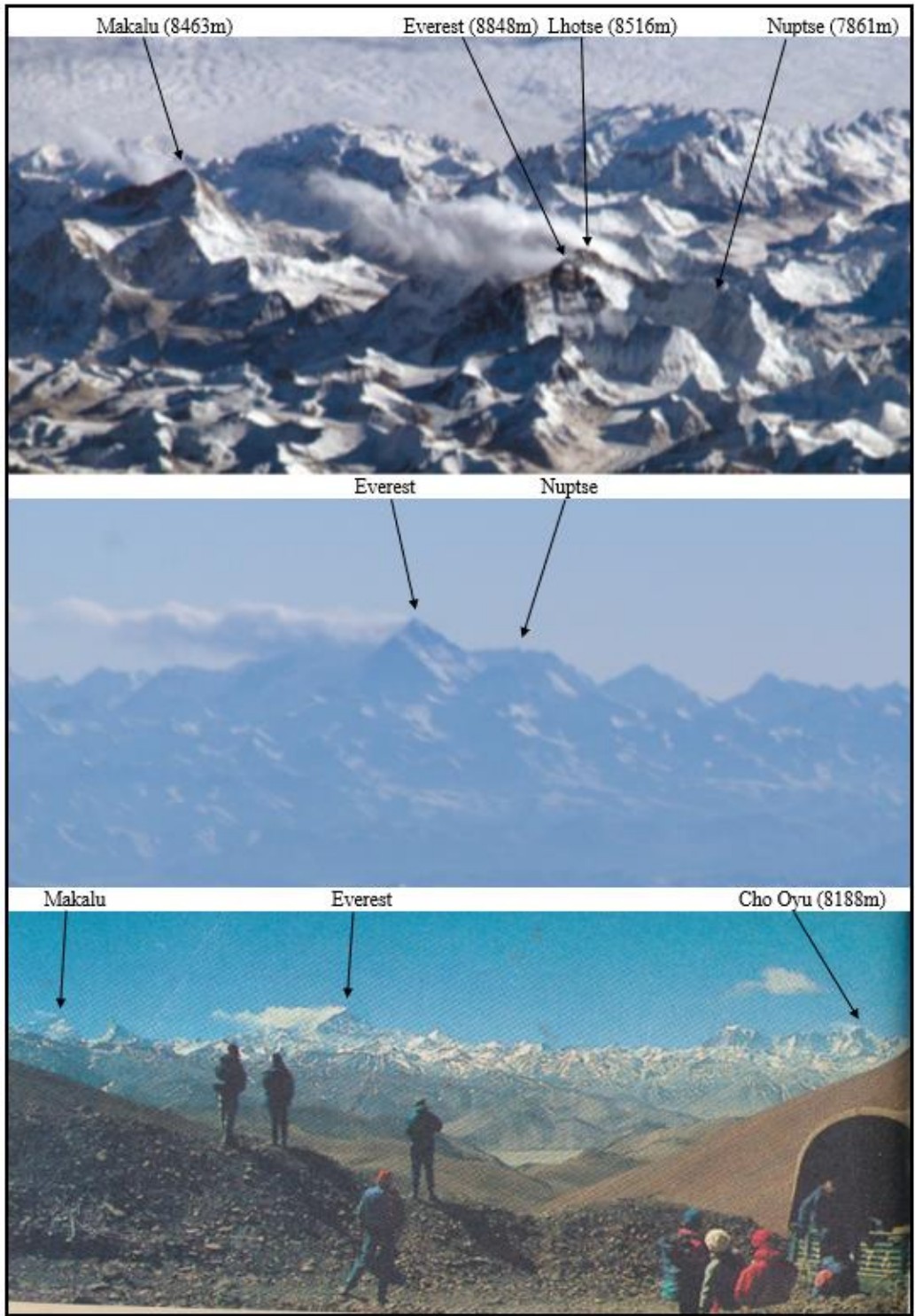

Figure 2. Top - The Everest plume studied by Moore (2004) imaged from the International Space Station (ISS) on 28 January
2004 at 1001 UTC (1601 LST, Local Solar Time).  Middle – The plume 3-minutes later from the ISS, not reported by Moore. Bottom - The Everest plume published in Venables (1989) photographed from the Pang La in Tibet on 6 March 1988 at about 0600 UTC (1200 LST).  The major peaks in the images are labelled and their summit elevations are given.

Plumes from the Everest massif were observed in November and December 1992 by Hindman and Engber (1995) as shown in Fig. 3 and captured in a video by Hindman in November 1995 (see Movie 1 in the Supplemental Material). As can be seen in the figure and in the video, the plumes were not present in the morning but were present in the afternoon. The video illustrates that the plumes formed like clouds and flowed and undulated like clouds. Based on this behaviour, plus investigations of the Everest airflow by Hindman and Wick (1990), Hindman and Engber reported these plumes were banner clouds.

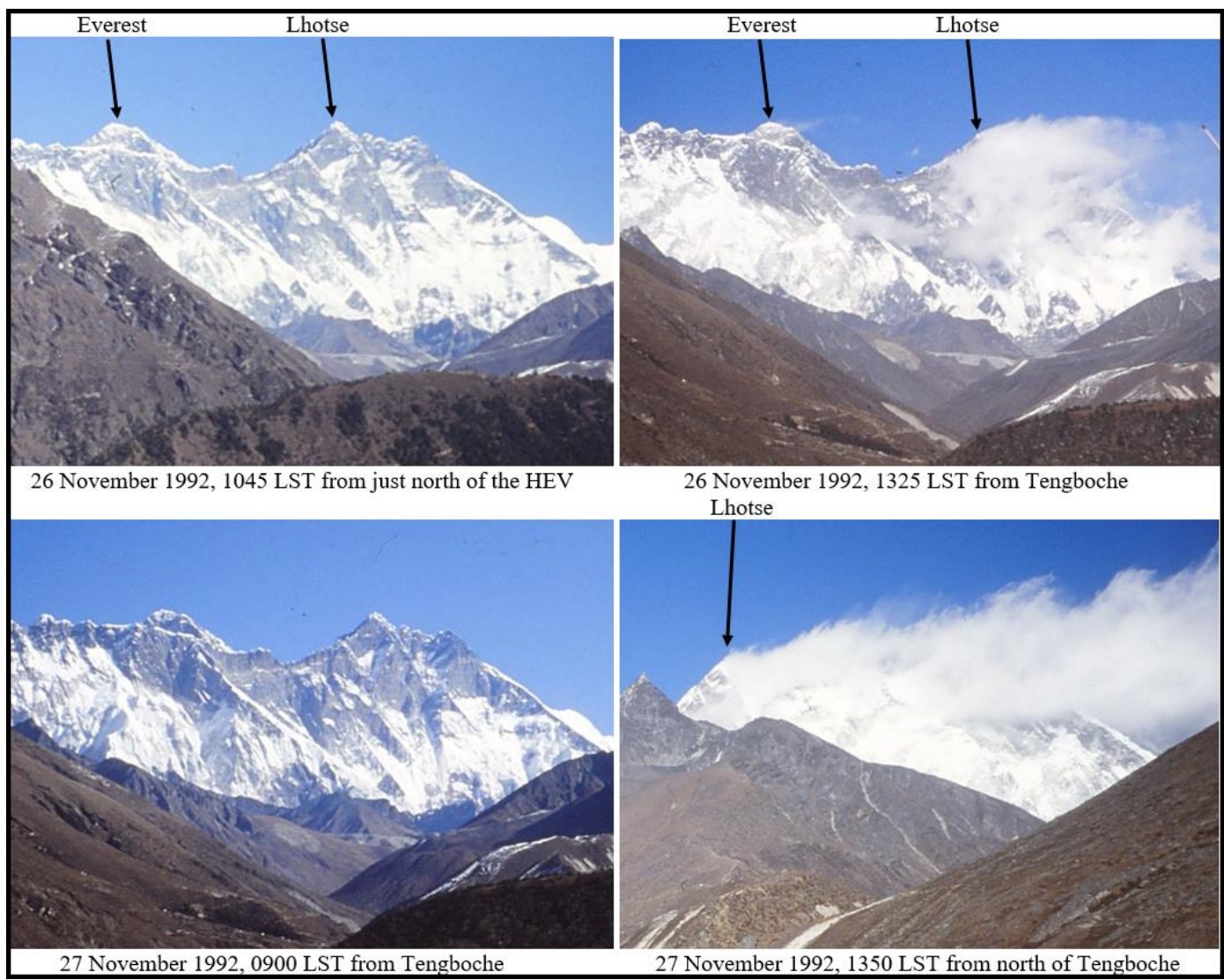

Figure 3. The plumes studied by Hindman and Engber (1995) photographed from the Nepal-side of the Everest massif during Hindman's trek to Everest's base.

Movie 1 captures the formation and evolution of a plume: The movie began at 0940 LST (Local Solar Time) showing the summits of Everest (poking over Nuptse) and Lhotse (to the right) were plume free. At about 1050 LST, a plume began in the wake of Lhotse. Clouds began to form on the valley slopes about 1200 LST. The plume reached full development at about 1400 LST. At that time, the plume began to be intermittently

obscured by clouds filling the valley. The movie ended at 1630 LST because the HEV was enveloped by the clouds that had completely filled the valley.

Numerical simulations by Reinert and Wirth (2009), Voigt and Wirth (2013) and Prestel and Wirth (2016) demonstrate banner clouds form in the lee of steep mountain peaks as a result of dynamically-forced lee upslope flow. This result confirms the flows postulated by Hindman and Wick (1990) that were inspired by Douglas (1928). The simulations show the speed of the lee upslope flow is much smaller than the speed of the wind impacting the peak. Thus, we think the lee upslope flow may be too weak to resuspend snow.

Schween and colleagues (2007) show still images and animations, all with the same view, from the summit of the Zugspitze in the Bavarian Alps. Because of the best possible spatial and temporal resolution, they were able to show the formation of banner clouds and snow blown off an adjacent peak.

Here we use the best possible spatial and temporal resolution images available to us from a geostationary meteorological satellite to observe the formation of plumes in the lee of the Everest massif. When we saw a plume form in the morning and if our calculations predicted cloud formation through condensation of moisture in the airstream upwelling in the immediate lee of the massif, the plume was likely a banner cloud. The composition of the cloud was inferred from its temperature.

**2 Procedures**

To our knowledge there is no systematic imaging of the Everest massif from either Nepal or Tibet (China). [Note: Anonymous reviewer (2022) informed us of a live-stream of the massif from the HEV (www.youtube.com/watch?v=RgDjOg4WvGI). The stream was not useful for this study because it began in January 2022)]. Therefore, daily we observed real-time, every ten-minute images of the Everest region during the 2020-21 winter (1 November through 31 March). Observed were Band 3 (visible) and Band 13 (infrared) from the Himawari-8 (H-8) Japanese geosynchronous meteorological satellite (www.data.jma.go.jp/mscweb/data/himawari/sat_img.php?area=ha2).

The spatial resolution of the H-8 images is sufficient to resolve the plumes, not as they form, but after they reach a length of a couple of kilometers. The following is our reasoning. The sub-satellite point is at 0 N, 104.7 E and the summit of Everest is at 27.99 N, 86.93 E. At the sub-satellite point, the satellite zenith angle is 0 degrees (nadir) and the spatial resolution is 0.5 km for images in the visible band and 2.0 km for images in the infrared band. Careful examination of pixel edges suggests that the 0.5 km and 2 km nadir resolutions increase to, respectively, about 1 km and 4 km in the vicinity of Everest. Moore (2004) estimated the plume

he studied, shown in Fig. 2, to be 15 km long.  Also comparing the plumes in Fig. 3 with the map in Fig. 1a, it can be seen that the plumes were kilometers in length.  So, had the H-8 been in orbit in 1992 and 2004, these plumes would have been observed.

The images from the H-8 website were displayed daily in the 'Hi-res Asia 2' window and observed in the both the 'still' and 'animation' modes.  The images could be magnified 300X on the FireFox browser and the site provided animations up to 23-hours before being overwritten.  The forming plumes were observed as moving elements against a mostly stationary background.  Once they reached a couple of kilometers in length, the lengthening of the plumes, shown in Movie 1, was observed.

To permit the reader to observe the formation and development of the plumes, we present movies made from the every-ten-minute H-8 images.  All of the H-8 images presented here are oriented such that the vertical points toward true north; Fig. 1c is a map of the region.  The map provides a distance scale and identifies the locations of the major peaks, the HEV, Phortse, Tengboche and the Arun Valley.  The times and dates for all the H-8 images are displayed on the images and the movies themselves.  The images and movies were produced following procedures in the Data Availability section.

Daily, we collected meteorological data corresponding to the H-8 images: atmospheric profiles (vertical distribution of temperature, moisture [dewpoint] and wind) from NOAA (www.ready.noaa.gov/index.php) at the location of Phortse, Nepal (27.84 N, 84.75 E, Fig. 1c); constant-pressure analyses of the region from the College of DuPage (weather.cod.edu/forecast/); surface measurements from the automatic weather station (AWS) at Phortse (earthpulse-raw.nationalgeographic.org/index.html).  The AWS is described by Perry, et. al. (2021).

Both Everest and its neighbor to the south, Lhotse, present significant obstacles to the typically west-to-east air flow (Fig. 1a).  Hence, both peaks produce wakes and, as seen in Fig. 2-top, both produced plumes.  Cloud formation was investigated in the dynamically-forced lee upslope flow in these wakes.  The lifted-condensation-level (LCL) of the upslope flow was calculated with the following procedure.

The atmospheric profiles were displayed using the American Skew-T adiabatic diagram. The profiles were graphically analysed to determine the LCL: the temperature and dew point values at the 400 mb level, the approximate pressure level at the base of the Everest pyramid, were raised, respectively, dry-adiabatically and with moisture constant to the level where saturation was achieved.  If the LCL was achieved before reaching the 300 mb level, the approximate pressure level at Everest's summit, a plume was expected to

form.  If the LCL was not achieved before reaching 300 mb, a plume was not expected to form; the unsaturated parcel would be swept downwind by the high-speed summit winds.  We checked the LCL values using www.csgnetwork.com/lclcalc.html.

The composition of a forming plume was inferred from the temperature at the LCL.  Baker and Lawson (2006) report the composition of mountain wave clouds, an analogue to the Everest plume.  They found the clouds could contain ice particles at temperatures colder than about -35 $^0$C.  Thus, if a LCL temperature was warmer than -35 $^0$C, initially liquid droplets are expected to have formed.  Conversely, if a LCL temperature was at or colder than -35 $^0$C, initially ice crystals are expected to have formed.  A mixed-phase plume (coexisting droplets and crystals) is expected near -35 $^0$C.

We looked for the following events in the daily H-8 images to identify the conditions in which plumes formed and the conditions in which plumes did not form:

1.  A day with no visible plume and no measured snowfall at Phortse either that day or the previous two days. This sequence will illustrate the H-8 view of the cloud-free Everest region and the corresponding non-plume atmospheric conditions.
2.  A day with a visible plume and no snowfall either that day or the previous two days at Phortse.  This sequence will illustrate the atmospheric conditions for plume formation.
3.  A day with a visible plume with no snowfall measured at Phortse that day but snowfall measured the previous three days, an event similar to Moore's (2004) study.  If the model does not predict a plume, we conclude the plume was composed of resuspended snow.  If a plume was predicted, we conclude the plume was a banner cloud.

We recorded the days the Everest massif was observed to produce a plume, the formation time of the plume, the plume duration and how many plume events were predicted by the LCL model.  Cases where a plume was observed but not predicted were investigated because they might have been plumes of resuspended snow.

We studied images from a geosynchronous meteorological satellite of the Moore (2004) plume event to determine if the plume behaved similarly as Event 3.

## 3 Results

### 3.1 Event 1

No plumes were observed (Figs. 4a, b and c) and no snowfall was measured at the AWS on 25, 26 and 27 January 2021.  Sharp-edge shadows cast by the Cho Oyu and Everest summits can be seen in these afternoon

images indicating no plumes were present. The shadows are more easily seen in Movie 2 for 2021-01-27. The movie begins just before sunrise and ends just after sunset, 0040 to 1150 UTC (0640 to 1750 LST). The Everest massif is in the center of the images. Scrolling slowly through the video, the long shadows in the morning cast by the massif can be seen shrinking and no plumes can be seen streaming from the summits. The shadows reappear in the afternoon. Further, the movie illustrates the snow-covered, cloud-free east face of Everest illuminated by the rising morning sun.

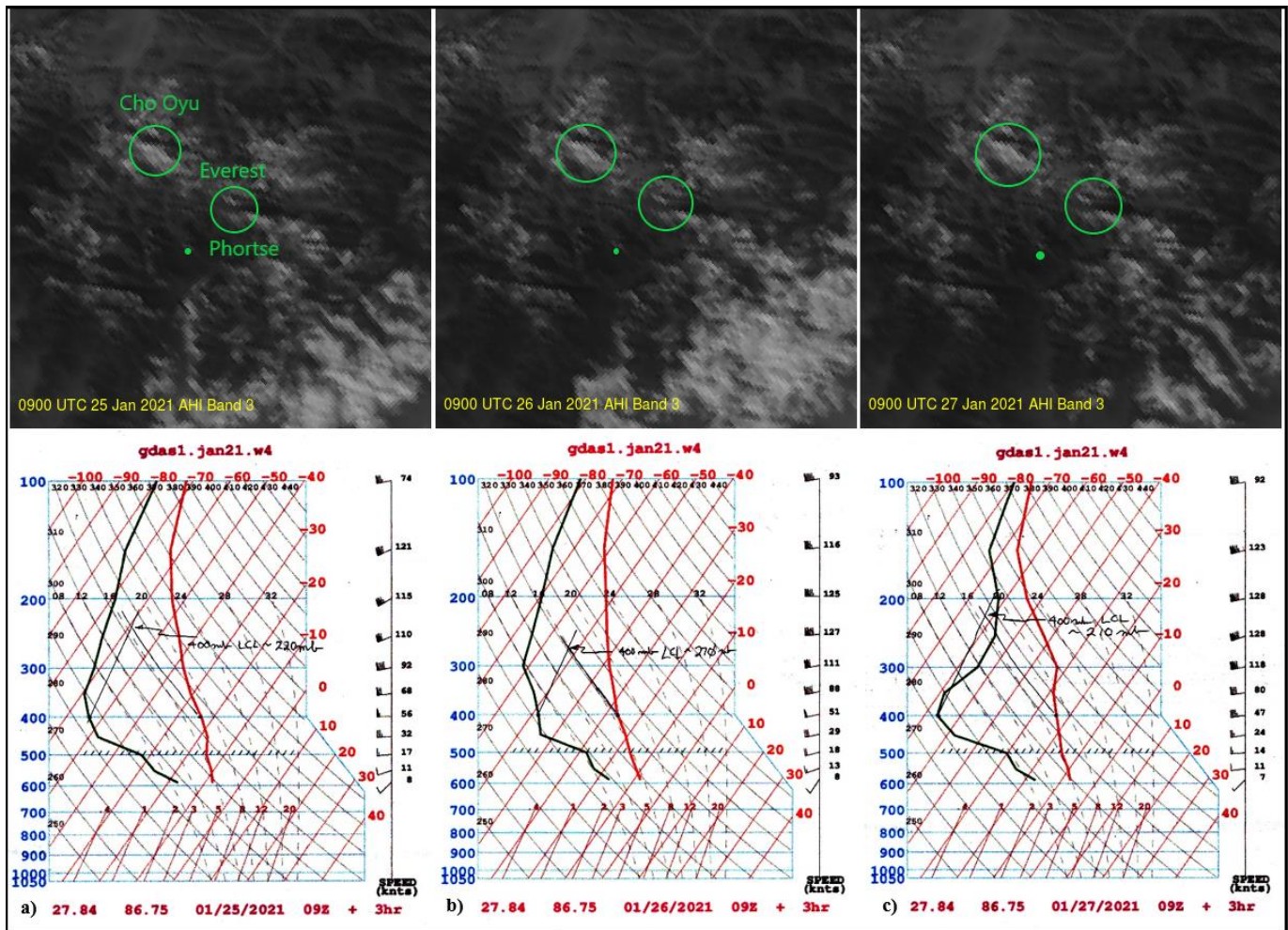

Figure 4: The images and profiles (a), (b), and (c) are for 2021-01-25, -26 and -27 at 15 LST or 09 UTC. The major peaks are circled and the location of Phortse is labelled. The lifting-condensation-level (LCL) values are determined graphically on the corresponding atmospheric profiles from Phortse and are listed in Table 1. The graphical procedures are described in the text. The pressures at the base and summit of the Everest pyramid, respectively, are approximately 400 and 300 mb.

We computed the LCL values, as illustrated in Fig. 4, on the atmospheric profiles corresponding to the images. The values are given in Table 1. It can be seen the values were all above the level of the Everest summit. The 400 mb levels were too dry. The temperature-minus-dew point ($T - T_d$) values were all 31 $^0$C or larger. This result is consistent with the observation of no plumes.

It can be seen from the profiles and in Table 1, the winds at the summit were from the west at about 100 knots (51 m/s) all three days.

Table 1

Air parcels lifted from the 400 mb level, approximate pressure at the base of the Everest summit pyramid, to their condensation levels (LCL) using *gdas1* profiles for Phortse, Nepal (27.84N, 84.75E). The approximate pressure at Everest's summit is 300 mb. The * signifies the IR images could not resolve a plume.

| Date | Time (LST) | Time (UTC) | T-Td at 400 mb (C) | LCL (mb) | T at LCL (C) | T at 300mb (C) | Plume expected? | Plume observed? | 300 mb winds (degrees/knots/m/s) |
|---|---|---|---|---|---|---|---|---|---|
| 25 Jan 2021 | 15 | 09 | 31 | 220 | -47 | -38 | No | No | 260/92/47 |
| 26 Jan 2021 | 15 | 09 | 33 | 270 | -48 | -27 | No | No | 260/111/57 |
| 27 Jan 2021 | 15 | 09 | 34 | 210 | -50 | -32 | No | No | 260/118/60 |
| 19 Dec 2020 | 15 | 09 | 23 | 280 | -42 | -37 | No | No | 290/103/53 |
| 20 Dec 2020 | 15 | 09 | 21 | 280 | -42 | -37 | No | No | 290/77/39 |
| 21 Dec 2020 | 15 | 09 | 4 | 380 | -27 | -38 | Yes | Yes | 270/81/41 |
| 8 Feb 2021 | 06 | 00 | 20 | 280 | -43 | -41 | No | No | 330/55/28 |
| 8 Feb 2021 | 09 | 03 | 15 | 290 | -40 | -39 | No | No | 330/60/31 |
| 8 Feb 2021 | 12 | 06 | 14 | 300 | -40 | -40 | Yes | Yes | 330/64/33 |
| 8 Feb 2021 | 15 | 09 | 13 | 310 | -39 | -40 | Yes | Yes | 330/70/36 |
| 8 Feb 2021 | 18 | 12 | 11 | 320 | -35 | -38 | Yes | Yes | 330/80/41 |
| 8 Feb 2021 | 21 | 15 | 10 | 330 | -34 | -37 | Yes | * | 320/82/42 |
| 8 Feb 2021 | 24 | 18 | 11 | 320 | -34 | -37 | Yes | * | 320/78/40 |
| 9 Feb 2021 | 03 | 21 | 13 | 310 | -35 | -36 | Yes | * | 330/86/44 |
| 9 Feb 2021 | 06 | 24 | 22 | 270 | -43 | -37 | No | No | 320/80/41 |

## 3.2 Event 2

A plume was observed on 21 December 2020 (Fig. 5c) but no snowfall was measured at the AWS between the 19[th] and 21[st]. As observed in Event 1, sharp-edge shadows cast by the Cho-Oyu and Everest summits in the 19[th] and 20[th] images (Fig. 5a and b) indicate no plumes were present. On the 21[st], plumes are seen streaming from these summits; the ovals in the image are elongated to bracket the plumes. Convective clouds are seen to the south of the peaks. These features are more easily observed in Movie 3 for 2020-12-21. The movie begins just before sunrise and ends just after sunset, 0040 to 1150 UTC (0640 to 1750 LST). Scrolling through the movie illustrates the late-morning onset of the plumes and convective clouds.

The LCL values computed on the profiles in Fig. 5 are given in Table 1. The values were above the level of the Everest summit the 19[th] and 20[th], consistent with the observation of no plumes. The 400 mb levels T-T$_d$ values were all 21 $^0$C or larger. The LCL value was below the summit level on the 21[st] consistent with the observed plumes. That 400 mb level T-T$_d$ value was 4 $^0$C, quite moist. The -27 $^0$C temperature at the LCL shows the plumes were likely liquid clouds.

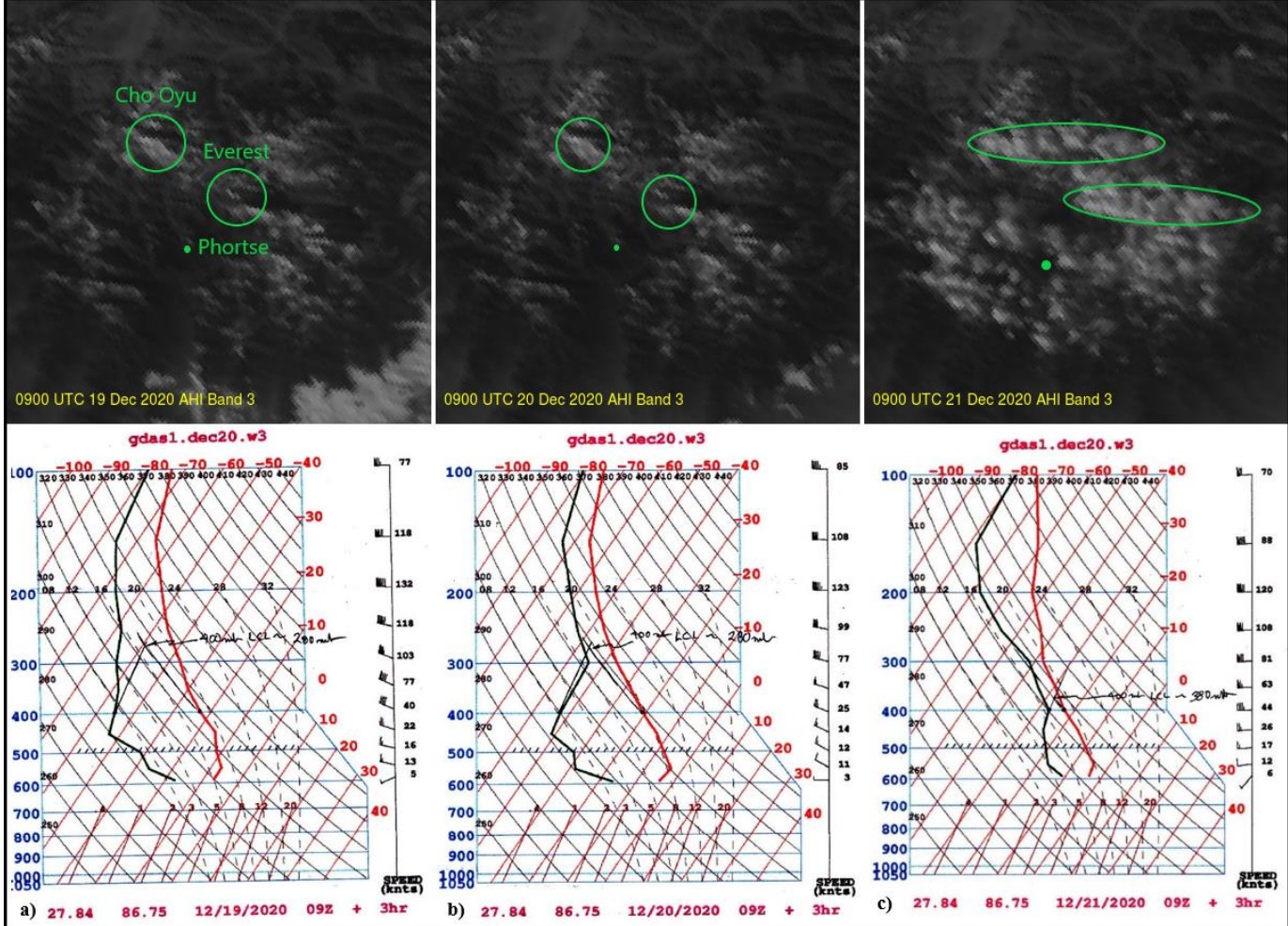

Figure 5. The images and profiles (a), (b) and (c) are for 2020-12-19, -20 and -21 at 09 UTC (15 LST). The major peaks are circled and the location of Phortse is labelled. The LCL values are determined graphically on the corresponding atmospheric profiles from Phortse and are listed in Table 1. The graphical procedures are described in the text. The pressures at the base and summit of the Everest pyramid, respectively, are approximately 400 and 300mb.

It can be seen from the profiles and in Table 1, the winds at the summit were from the west-north-west between 77 and 103 knots (39 and 53 m/s) for the three days.

### 3.3 Event 3

A plume was observed on 8 February 2021 (Figs. 6c, d and e) and snowfall was measured at the AWS on

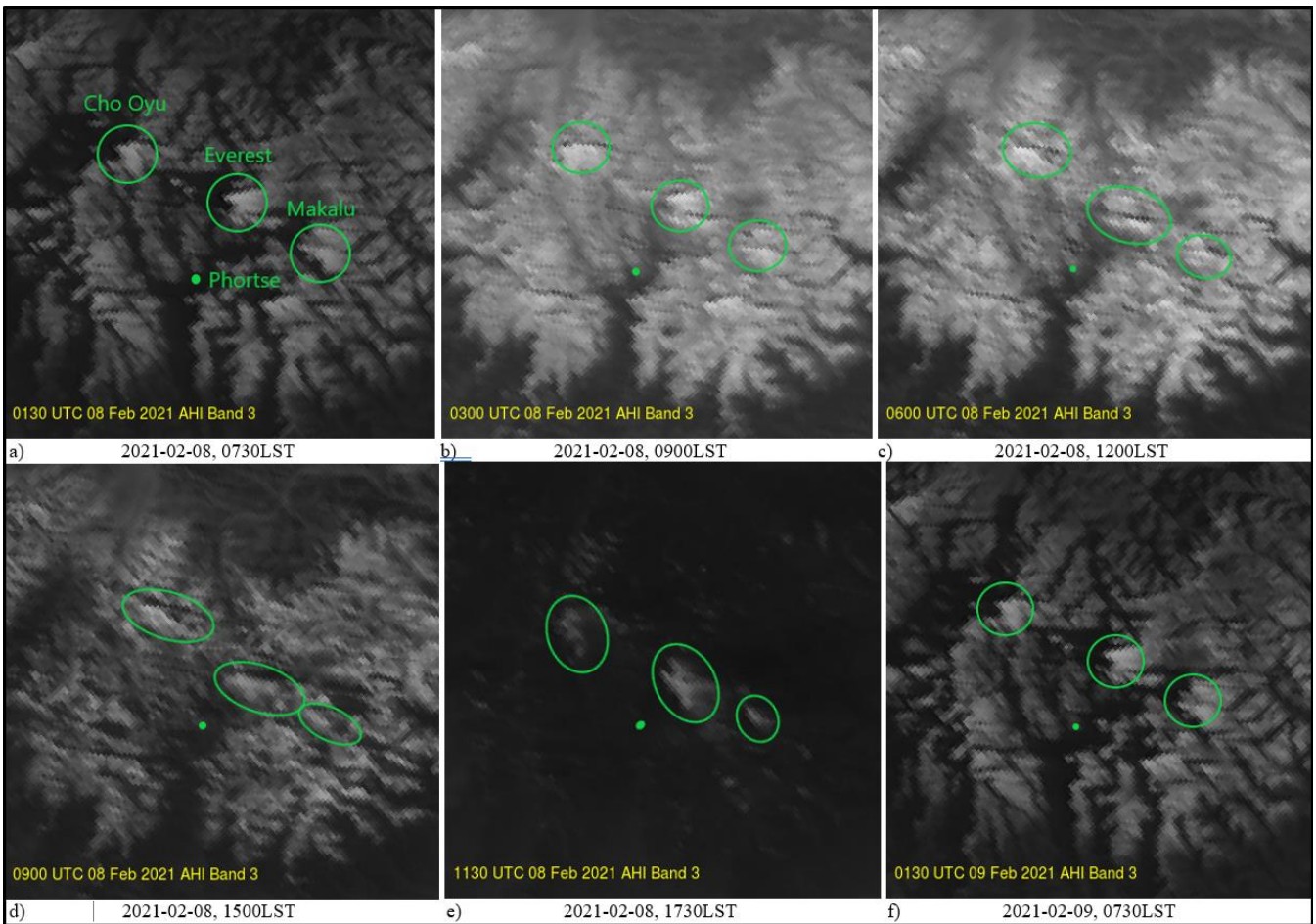

Figure 6. The visible images (a), (b), (c), (d) and (e) are for 2021-02-08 and (f) is for 2021-02-09 at LST (UTC + 6h). The locations of the major peaks are circled. The corresponding LCL values are in Table 1.

the 5[th] and 6[th] but none on the 7[th] and 8[th] (images from the 5[th] through 7[th] are not presented in Fig. 6 because the region was obscured by clouds from a passing Western Disturbance (Lang and Barros, 2004)). As can be seen in Figs. 6a and b, on the 8[th] shadows from the summits appear in the 0730 and 0900 LST images, indicating no plumes. Cho Oyu and Everest are producing plumes in the 1200 and 1500 LST images (Figs. 6c and d). These plumes along with Lhotse's and Makalu's plume are seen as the bright objects in the 1730 LST image (Fig. 6e). The corresponding 1730 LST infrared image did not resolve the plumes nor did the overnight infrared images. But, the visible image the next morning (Fig. 6f), the morning of the 9[th] at 0730 LST, is almost identical to the previous morning's image (Fig. 6a). Thus, the plumes dissipated overnight. No plumes were present either morning.

Features in Fig. 6 are more easily viewed in Movie 4 for 2021-02-08. The movie begins just before sunrise and ends just after sunset, 0050 to 1210 UTC (0650 to 1810 LST). Slowing the video using the scroll bar, the animation illustrates the development of the plumes in the afternoon and their final illumination at sunset.

At sunset, the movie reveals four plumes, one streaming from Cho Oyu's summit, a second from Everest's summit, a third from the summit of nearby Lhotse and the fourth from Makalu. The movie illustrates the plume from Lhotse was much larger than the plume from Everest.

The LCL values, shown in the Table 1, were above the level of Everest's summit (~300 mb) at 00 and 03 UTC (06 and 09 LST) consistent with the observation of no plumes. The LCL values were at and below the summit level between 06 and 12 UTC (12 and 18 LST) consistent with the observed plumes. The values remained below the summit level overnight. The next day the 24 UTC (06 LST) value is above the summit level consistent with the observation of no plumes.

It can be seen from Table 1, the winds at the summit were from the northwest between 55 and 86 knots (28 and 44 m/s) on the 8th and 9th. These winds were caused by the jet-stream that moved through the Everest region during the 8th and 9th as shown by the sequence of images in Fig. 7. The red sinuous region defines the jet stream. Additionally, it can be seen in the sequence the trough of the Western Disturbance, in which 235 the jet stream was embedded, was east of the Everest region and was moving slowly eastward.

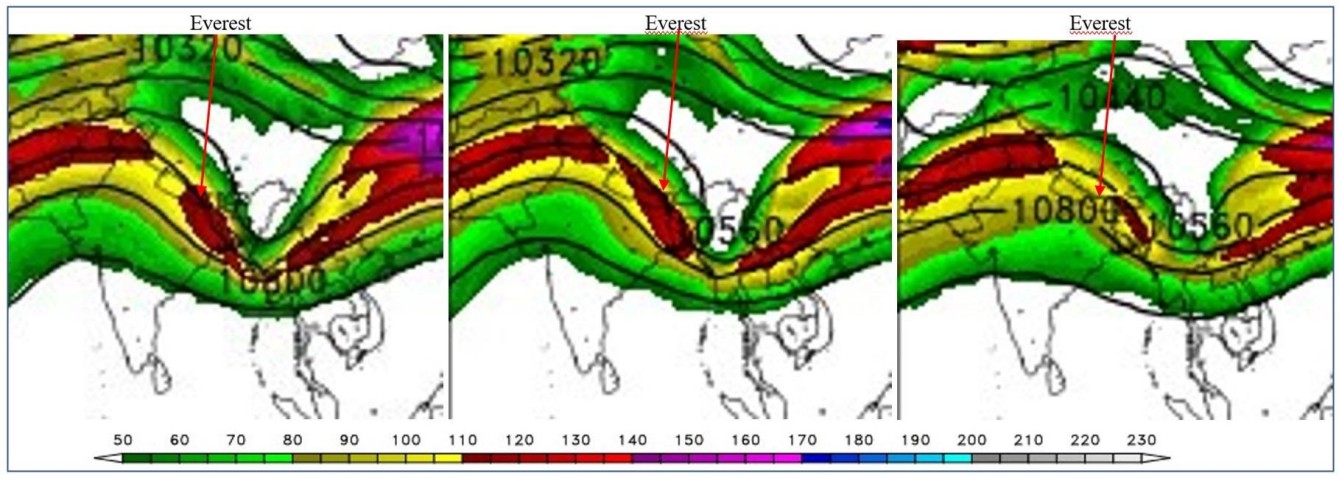

Figure 7. The 00 UTC Global Forecast System forecast for 8 February 2021: Left, 00 UTC (06 LST); Center: 12 UTC (18 LST); Right, 24 UTC (06 LST 9 Feb.). Shown are the 250 mb isotachs (knots) in the colour scale, geopotential heights (gpm) and the location of Everest. Collected from the College of DuPage NEXLAB website.

**3.4 Plume statistics**

Table 2 displays a summary of our daily observations of the H-8 imagery and the 400 mb LCL values calculated from the corresponding atmospheric profiles. It can be seen from the table, Everest was almost always visible, 143 of the 151 days (95%). On the days Everest was visible, plumes were observed to form on 63 days (44%). Of these plumes, 59 (94%) were predicted to form and 4 (6%) were not predicted. Were 245 those four plumes composed of resuspended snow?

Table 2

Results from the observations of Himaeari-8 imagery and the Lifted Condensation Level (LCL) calculations from the corresponding atmospheric profiles (yellow shading denotes summed values, green shading denotes average values)

| Month 2020-21 | Number of days observed | Everest visible | Plume observed | Average plume formation time (hour LST) | Average plume duration (hours) | Average LCL temperature ($^0$C) | Aver age 300 mb winds (degrees) | (m/s) | Plume predicted | Plume not predicted |
|---|---|---|---|---|---|---|---|---|---|---|
| November | 30 | 26 | 7 | 10 | 8 | -32 | 271 | 38 | 7 | 0 |
| December | 31 | 31 | 15 | 7 | 14 | -31 | 268 | 47 | 14 | 1 |
| January | 31 | 31 | 7 | 9 | 14 | -31 | 266 | 45 | 6 | 1 |
| February | 28 | 28 | 17 | 9 | 11 | -35 | 247 | 28 | 15 | 2 |
| March | 31 | 27 | 17 | 9 | 11 | -34 | 269 | 32 | 17 | 0 |
| Sum | 151 | 143 | 63 | 9 | 12 | -33 | 264 | 38 | 59 | 4 |
| | | 95% | 44% | | | | | | 94% | 6% |

The four plumes were observed on 2020-12-05, 2021-01-29 and 2021-02-03 and 2021-02-11. The 400 mb LCL values for the plumes ranged from 295 to 249 mb, all above the 300 mb level of the Everest summit. The plumes formed between 1200 and 1400 LST and dissipated around 1900 LST. The plumes were not visible at sunrise. Therefore, these plumes were not composed of resuspended snow. Thus, none of the 63 plumes we observed were composed of resuspended snow. However, plumes of resuspended snow may have occurred smaller than the H-8 detection limit of a couple of kilometers.

Twice-daily images of the Everest summit coincident with a portion of our H-8 observations became available from Grey, et al. (2022) while this study was in peer-review. The images were taken from 2020-12-16 through 2021-01-16 (32 days) at ~10 and ~17 LST. We studied the images to determine the number of days the summit was visible and the number of days plumes occurred. The summit was visible on 28 days (88%) while the corresponding H-8 observations revealed the massif was visible on 32 days (100%). The summit produced 18 morning plumes and 11 afternoon plumes. The corresponding H-8 observations detected 8 of the morning plumes and 4 of the afternoon plumes. This comparison shows a number of Everest plumes did not reach the couple-of-kilometers in length to be detected in the real-time H-8 images.

We observed plumes we suspect were composed primarily of snow formed in situ as shown in Movie 5. The movie was constructed from the real-time H-8 infrared images as described in the Data Availability section. Note, in the movie NT is Nepal Time which is approximately Local Solar Time. The major summits are labelled and are seen as white, stationary objects. On 2020-12-21 plumes are seen to form in the morning downwind of the Everest massif and Cho Oyu (also, these plumes are shown in Movie 3). The plumes dissipated four days later on 2020-12-25 early in the morning. The plumes fluctuated in length and can be seen to stream well into Tibet. The 400 mb LCL values were between 393 and 356 mb, extremely moist conditions, though no precipitation was measured at Phortse.

**3.5 The Moore plume**

Moore (2004) studied plumes that streamed from Everest-Lhotse-Nuptse massif late in the afternoon of 28 January 2004 (Fig. 2-top and middle). The plumes were imaged from the International Space Station (ISS). To determine if the plumes were present that morning and the next, we analysed all available images from the Geosynchronous Orbiting Environmental Satellite-9 (GOES-9). The GOES-9 was lent by the USA to Japan after the failed launch of MTSAT-1.

The GOES-9 images are shown in Fig. 8. The early-morning image at 0725 LST on 28 January (Fig. 8a) shows sharp-edge shadows from Everest and Makalu. Had the plumes been present, the shadows would have been fuzzy and diffuse. The plumes were not visible until lit by the late afternoon sun as seen in the 1613 and 1649 LST images (Figs. 8c and d). This illumination of the plumes at sunset also occurred for the plumes

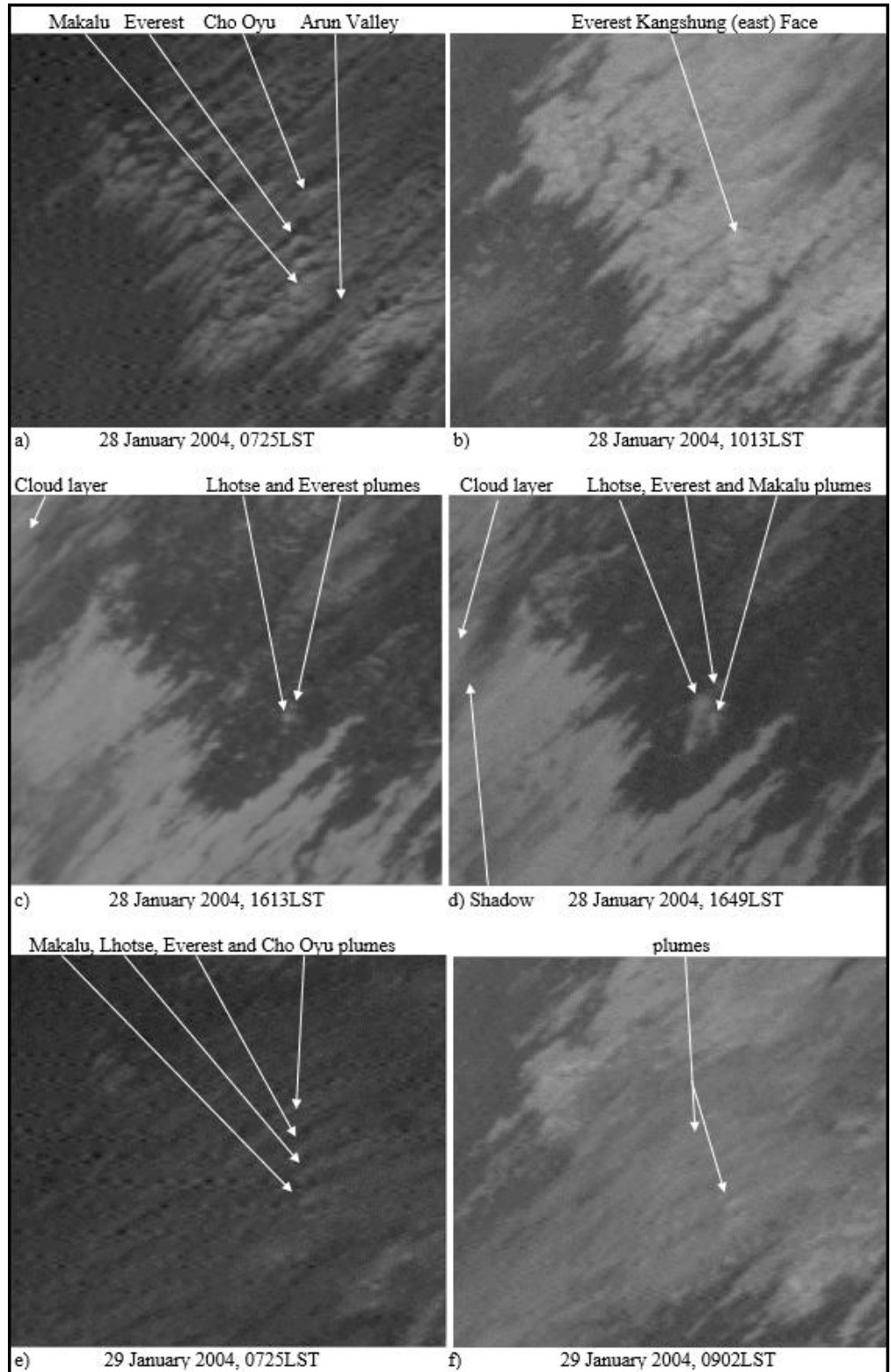

Figure 8. GOES-9 0.65 micrometer images of the study region. The major features are labelled.

The GOES images for the afternoon of 28 January show a cloud layer moved toward the Everest region from the west. The layer is visible in the 1613 and 1649 LST images (Figs. 8c and d). In the 1649 LST image,

the layer cast a shadow on the lower clouds. Moisture ahead this layer may have formed the afternoon plumes imaged from the ISS. Based on this interpretation of the GOES images, we conclude the plume Moore studied was not present in the morning and formed in the afternoon.

Overnight, the cloud layer moved into the Everest region because at dawn on 29 January, the plumes produced by the major summits are seen to protrude above the overcast (0725 and 0902 LST images, Figs. 8e and f). The protruding plumes are difficult to identify in the figures. So, we searched the archives for finer spatial-resolution images from polar-orbiting satellites.

Finer detail of these plumes was found in the Terra/MODerate resolution Imaging Spectroradiometer (MODIS) visible image of 0910 LST on 29 January 2004 (Fig. 9). The spatial resolution of this MODIS image is 0.38 km per pixel: the distance between Everest and Lhotse summits is 3 km (Fig. 1a) and 8 pixels cover that distance. Unfortunately, the MODIS visible image on the 28[th] was not useful because it was on the limb and pixilated, smearing the features. The MODIS 0.85 micrometer wavelength image is good for cloud detection (compared to 0.65 micrometers on GOES) because atmospheric scattering is less at 0.86 micrometers and contrasts are better maintained.

The MODIS image shows distinct plumes in the wakes of the major peaks. The Everest plume casts a shadow on the lower cloud layer indicating it rises above that layer. The shadow indicates the plume has a sharp edge, the edge of a liquid cloud. A short distance downwind, the plume merges with the plume from Lhotse and becomes fuzzy, suggesting glaciation. The regions of the plumes containing primarily cloud droplets are the most reflective hence the brightest, the whitest. The region of the plume containing primarily much larger ice crystals are less reflective and appear dimmer and grayer. The fuzzy plume traveled across the Arun Valley. It is possible crystals fell as snow that may have reached the surface.

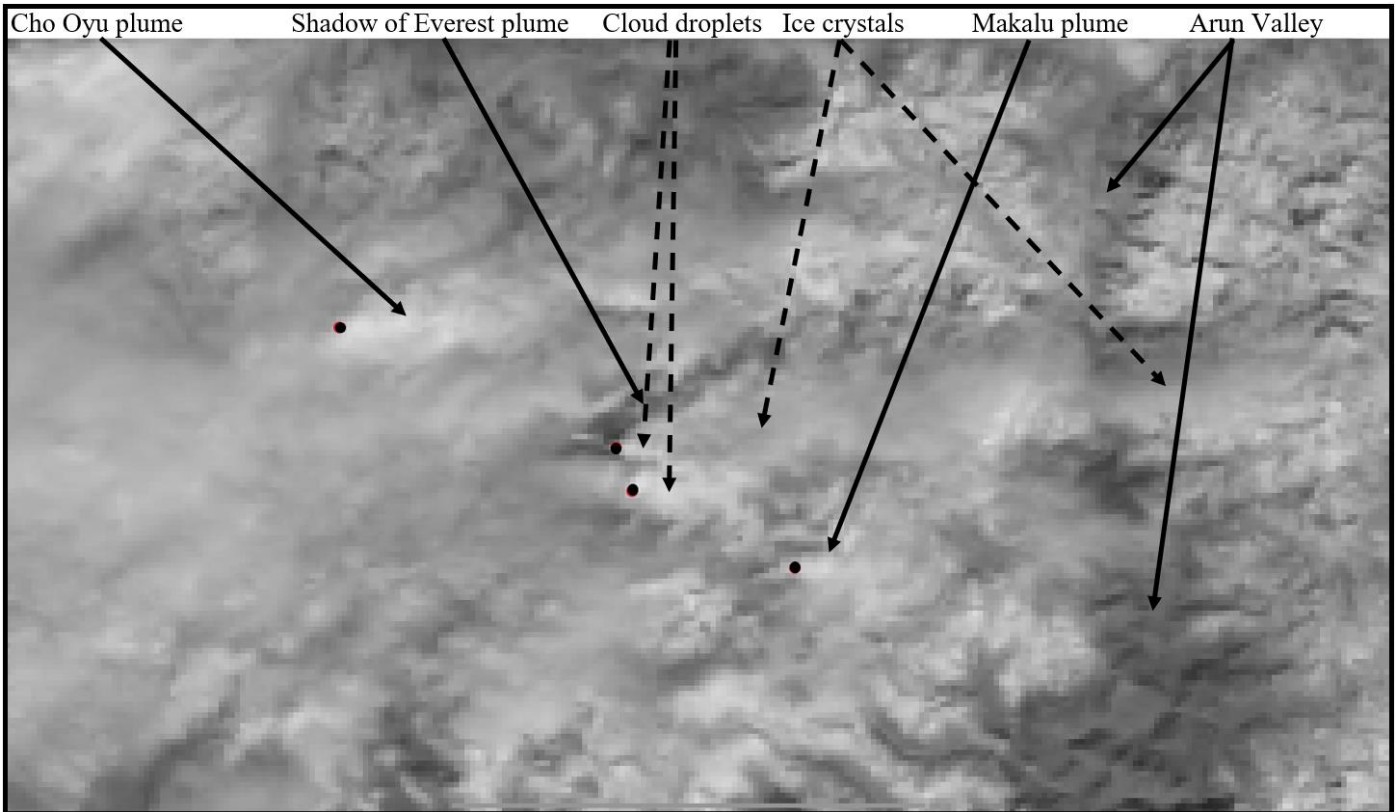

Figure 9. Terra/MODIS 0.85 micrometer visible image of the study region on 2004-01-29 at 0910 LST. The main features are labelled. The black dots are the locations of the summits. Figure 1c is a map of the region displayed in this image.

## 4 Discussion

### 4.1 Meteorology

The plume observations and the corresponding meteorological analyses are summarized in Tables 1 and 2. The LCL values show plumes were observed when the 400 mb LCL was below the 300 mb level of the summit of Everest. This result shows that moisture condensed in the dynamically-forced rising air in the Everest wake to produce the plumes. Moisture likely was transported vertically in morning convection (Hindman and Upadhyay, 2002) and entrained by the wake producing the afternoon plumes. Some of the moisture could have come from sublimation of snow. Stigter, et al. (2018) measured cumulative sublimation and evaporation from a glacier in the Nepalese Himalayas to be 21% of the total annual snowfall. Finally, the morning moisture transport and afternoon appearance of the plumes are consistent with the findings of Wirth, et al. (2012, Fig. 5b) for banner clouds produced by Mount Zugspitze.

All the plumes we presented (Figs. 3, 5 and 6) were absent in the mornings and visible in the afternoons. The plumes with corresponding meteorology (Figs. 5 and 6) occurred with summit wind speeds 50 knots (26 m/s) or greater and 400 mb $T-T_d$ values of 14 $^0$C or less. If the $T-T_d$ values were larger than 14 $^0$C, no plumes were observed.

The plume Moore (2004) investigated was not observed in the morning (Figs. 8a and b). Had it been a plume of resuspended snow, as he concluded, the plume would have been visible in the morning because the wind speeds were between 85 and 120 knots (43 and 61 m/s) all day (from the REANALYSIS archive at www.ready.noaa.gov/READYamet.php). On the next day, the plume was observed in a MODIS image to glaciate downwind (Fig. 9). Thus, the plume may have produced snow.

**4.2 Composition**

The initial composition of the plumes was deduced from the temperature of the LCL. The initial composition of the 21 December 2020 plumes (Fig. 5c) was expected to be cloud droplets because the plume formed at a temperature warmer than -35 $^0$C. The plumes of 8 February 2021 (Figs. 6d and e) likely began as ice clouds because the plumes formed at a temperature colder than -35 $^0$C. The Everest plume imaged in Fig. 9 appears initially liquid that glaciated downwind. This change in composition is supported by the measurements of Baker and Lawson (2006) that revealed cloud droplets that formed initially could nucleate to form ice/snow crystals further downwind (their Fig. 6).

The plumes we observed, plus Moore's, could not have been composed of resuspended snow because they were not present in the mornings. The wind speeds were fast from morning throughout the day. If the plumes were composed of resuspended snow, they also would have appeared in the mornings.

**4.3 Estimate of snowfall from the observed plumes**

Assume a saturated parcel of air ascends moist adiabatically in Everest's wake from the elevation of the South Col at ~7900 m (~400 mb) to the summit at ~8900 m (~300 mb), see Fig. 1a. The parcel is initially -33 $^0$C (the average plume temperature, Table 2) and cools to -40 $^0$C at the summit. The initial parcel saturated mixing ratio is 0.59 g/kg and the final is 0.39 g/kg for an average of 0.49 g/kg. Employing the precipitable water calculator at www.shodor.org/os411/courses/_master/tools/calculators/precipwater/, ~1 mm of water is expected to precipitate from the parcel.

Assume the parcel ascends at 0.1 m/s in the turbulent wake the 1000 m from the South Col to the summit, the ascent requires $10^4$ s. So, every $10^4$ seconds 1 mm of liquid precipitates from the parcel. The average duration of the observed plumes was 12 hours (Table 2) or 4.32 x $10^4$ seconds. The amount of precipitation from the average plume was 1 mm/$10^4$ s x 4.32 x $10^4$ s or about 4 mm.

Sixty-three (63) Everest plumes occurred during our four-month observation period (Table 2). So, 63 plumes x 4 mm/plume equals about 252 mm (~25 cm) of liquid-equivalent. The amount of liquid-equivalent

precipitation measured at Phortse during our observation period was 284.5 mm (~28 cm). Thus, Everest plumes may be a significant source of snowfall.

The plume-generated snowfall is expected to be a maximum in the immediate lee of the Everest massif and
diminish downwind as drier air is entrained. The always-white Kangshung face of Everest (Fig. 1b) may be evidence of plume-generated snowfall, although, much of this snow may be captured from snow-filled clouds flowing around the summit pyramid. This capture is similar to snow collecting on the tailgate of a truck speeding through a snow storm.

**5 Conclusions**
We studied the formation and composition of wintertime plumes produced by the Mt. Everest massif. We found the massif produced plumes when the air entrained into its wake was sufficiently moist, 400 mb temperature-minus-dew point values 14 $^0$C or smaller. The plumes occurred with summit winds 50 knots (26 m/s) or greater. We concluded one plume initially was composed of cloud droplets, not resuspended
snow, and the other was initially composed of ice particles. Evidence is presented that one plume glaciated downwind. We estimated the plumes may be a significant source of snowfall.

The Everest massif was visible on 143 of the 151 observation days (95%) especially in the morning because the plumes most often formed later in the morning. On the days the massif was visible, plumes were observed
to form on 63 days (44%). The plumes lasted an average of 12 hours. Of these plumes, 59 (94%) were predicted to form and 4 (6%) were not predicted. These four plumes were not composed of resuspended snow because they were not visible at sunrise. However, plumes of resuspended snow may have occurred smaller than the couple of kilometers detection limit of the Himawari-8 satellite for the Everest region.

Our analysis of the Grey et. al. (2022) images of the Everest summit from the surface showed a number of Everest plumes did not reach the couple of kilometers in length to be detected in the real-time H-8 images. Thus, our plume-occurrence values should be considered a lower-limit.

The plume studied by Moore (2004) we show was a banner cloud, not a plume of resuspended snow.

**Data availability**
The images in Figs. 4, 5 and 6 were created using Geo2Grid software (cimss.ssec.wisc.edu/csppgeo/geo2grid_v1.0.0.html) and Himawari Standard Data (HSD) files from H-8 available at the UW-Madison SSEC Data Center (courtesy of JMA, the Japan Meteorological Agency).


Movies 2, 3 and 4 were created from the still imagery using ImageMagick. Tutorials on how to use Geo2Grid are available at this CIMSS Satellite Blog link: cimss.ssec.wisc.edu/satellite-blog/?s=geo2grid. The movies, themselves, are in the accompanying Supplemental Material.

Movie 5 was constructed from *.GIF images downloaded in real-time from the Himaware-8 website (www.data.jma.go.jp/mscweb/data/himawari/sat_img.php?area=ha2). Images were downloaded every 30 minutes. The images were animated and labelled using EzGIF.com and the animation was transformed to MP4 using VideoPad Video Editor.

Wirth (2022) suggested we attempt to post-process the best-resolution H-8 visible imagery to improve the movie resolution. In general, the sharpening techniques we are aware of (in SatPy for example) require a higher resolution band. So, for example on H-8, Band 1 (0.47 micrometers, with 1-km resolution at nadir) or Band 2 (0.51 micrometers, also 1 km resolution) can be sharpened with information from Band 3 (0.64 micrometer, with 0.5 km resolution at nadir). So, there is no practical method to improve the spatial 415 resolution in Band 3.

Data for the MODIS imagery were downloaded from the NASA LAADS (Level-1 and Atmosphere Archive and Distribution System) DAAC (Distributed Active Archive Center) archive and processed into imagery using Polar2Grid software available at (www.ssec.wisc.edu/software/polar2grid/). A tutorial on how to 420 access and display archived MODIS data is here: cimss.ssec.wisc.edu/satellite-blog/archives/36727.

**Author contributions**

Edward Hindman initiated the study, provided the meteorological interpretations and constructed Movies 1 and 5. Scott Lindstrom produced the satellite images, Movies 2, 3 and 4 and satellite sensor interpretations.


**Competing interests**

The authors declare that they have no conflict of interest.

**Acknowledgements**

EH's participation was self-funded. SL was partially supported by the Cooperative Agreement between NOAA and the University of Wisconsin-Madison Cooperative Institute for Meteorological Satellite Studies (CIMSS).

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
