# Peer review of "The formation and composition of the Mount Everest plume in winter"

_Atmospheric Chemistry and Physics, 2021_

## Author Comment (AC2)

[Figure]

Figure 6. The visible images are for 2021-02-08 and -09 at Local Solar Time (LST). The locations of the major peaks are circled. The corresponding LCL values are in Table 1.

---

## Author Comment (AC5)

**RC2**: 'Comment on acp-2021-966', Anonymous Referee #2, 10 Feb 2022:

The manuscript "The Mount Everest plume in Winter" by E. E. Hindman and S. Lindstrom analyses the formation and composition of the Mount Everest plume. To do so, the authors examined two wintertime plumes in detail by using GMS images and atmospheric sounding data, which were used for an atmospheric model simulation. The results of this work conclude that the identified plumes are either composed of cloud droplets or ice particles, but were not composed of resuspended snow.

The authors describe their analysis method very clearly and summarize the history of plume studies at Mount Everest in a very nice way. Their analysis method is simple, but was never done before for the Mount Everest plume. For this reason, I recommend to publish this work in ACP after a major revision.

*We thank this reviewer for their careful and thorough reading of our manuscript. We respond to their comments in italics:*

**Major comments**:

This study is mainly based on satellite images with a very low resolution. For me as a reader, it was very hard to see the described observations on the images and especially the videos. I don't doubt their method, but I think it is worth it to put some more effort in the presentation of the images and videos to make the authors' statements clearer. I would recommend to add labels with time, km scale bars and markers for the location of the summits.

*In response to this comment, the following paragraph will be inserted between lines 77 and 79:*

>*To locate the features and know distances in the H-8 images and movies, a map is given in Figure 1a. The map provides a distance scale and identifies the locations of the major mountain peaks, the HEV, Phortse and the Arun Valley. The times and dates for all the H-8 images are displayed on the images and the movies themselves.*

*Figure 1 will be expanded to illustrate the important features of the Mt. Everest region presented in the manuscript:*

[Figure]

*Figure 1. **a**. The Mount Everest region with the major summits and locations identified; the HEV is the Hotel Everest View. **b**. The Mount Everest and Lhotse summit pyramids are identified, respectively, by the black-dashed and black dash- dot lines. The bases of the pyramids are at an elevation of approximately 7900m. The summits are, respectively, 8848m and 8501m in elevation. The map segment is from the November 1988 issue of the National Geographic Magazine. **c**. The Everest summit pyramid at sunrise in May 2010 as viewed from near the summit of Lhotse (from CoryRichards.com and Anker, et. al. (2013)).*

*The Anker, et. al. reference will be added to the list: Conrad, A., et. al., The call of Everest. National Geographic Society, 303 pp. ISBN 978-1-4262-1016-7, 2013.*

The authors are focusing on the development process of the plumes and describe that the observed plumes and the Moore plume are not composed of resuspended snow. It gives the impression that plumes from resuspended snow are not possible at all, which I don't think the authors intend. I think the Authors should make clear that this is just a case study over a few days (see Table 1) and plumes of resuspended snow might be still possible.

*In response to this insightful comment, the following text will be inserted between line 111 and 113:*

> *All our daily observations were analyzed. Recorded were the days the Everest massif was observed to produce a plume, the formation time of the plume, the plume duration and how many plume events were predicted by the LCL model. Cases where a plume was observed but not predicted are investigated because they might be plumes of resuspended snow.*

I would also be interested in some more statistics. For example, how often do these plumes occur per month and how long do they usually persist?

*To answer this important suggestion, a new section will be inserted before Section 3.4:*

**3.4 Plume statistics**

*Table 2 displays the results from our 151 daily observations of H-8 imagery and the corresponding 400 mb LCL values calculated from the atmospheric profiles. It can be seen from the table, Everest was almost always visible (95%), especially*

| | | | | | | | | | | |
|---|---|---|---|---|---|---|---|---|---|---|
| **Table 2** | | | | | | | | | | |
| Results from the observations of H-8 imagery and Lifted Condensation Level (LCL) calculations from the corresponding atmospheric profiles | | | | | | | | | | |
| Month 2020-21 | Number of days observed | Everest visible | Plume observed | Average plume formation time (hour LST) | Average plume duration (hours) | Average LCL temperature ($^0$C) | Aver age 300 mb (degrees) | winds (m/s) | Plume predicted | Plume not predicted |
| November | 30 | 26 | 7 | 10 | 8 | -32 | 271 | 38 | 7 | 0 |
| December | 31 | 31 | 15 | 7 | 14 | -31 | 268 | 47 | 14 | 1 |
| January | 31 | 31 | 7 | 9 | 14 | -31 | 266 | 45 | 6 | 1 |
| February | 28 | 28 | 17 | 9 | 11 | -35 | 247 | 28 | 15 | 2 |
| March | 31 | 27 | 17 | 9 | 11 | -34 | 269 | 32 | 17 | 0 |
| Sum | 151 | 143 | 63 | 9 | 12 | -33 | 264 | 38 | 59 | 4 |
| | | 95% | 44% | | | | | | 94% | 6% |

*in the morning because the plumes most often formed later in the morning. On almost half of the days Everest was visible (143 days), plumes were observed to form on 63 days (44%). Of these plumes, 59 (94%) were predicted to form and 4 (6%) not predicted. Were the four plumes composed of resuspended snow?*

*Here are the steps we used to determine if a plume was a banner cloud or resuspended snow. Scrolling frame-by frame through the Movie 4 (Event 3) shows the plume was not visible as the sun rose and was clearly visible as the sun set. This behavior was consistent with the LCL results: the LCL was above the Everest summit in the morning and below the summit in the afternoon; this plume was a banner cloud. If this plume had been visible at sunrise and the corresponding LCL value was above the Everest summit, then the plume was composed of resuspended snow. These steps are applied to the four plumes suspected to be blowing snow.*

*The four plumes were observed on 2020-12-05, 2021-01-29 and 2021-02-03 and 11. The 400 mb LCL values ranged between 295 to 249 mb, all above the 300 mb level of the Everest summit. The plumes formed between 1200 and 1400 LST and dissipated around 1900 LST. The plumes were not visible at sunrise and visible at sunset. Therefore, these plumes were not composed of resuspended snow. Thus, none of the 63 plumes we observed we conclude were composed of resuspended snow. Though, plumes of resuspended snow may have occurred smaller than our detection limit of a couple of kilometers.*

*We observed, though, plumes we think were composed primarily snow formed insitu. For example, on 2020-12-20, plumes were observed in the H-8 infrared images to form downwind of the Everest massif at 1100 LST and dissipate four days later on 2020-12-24 at 1900LST. For those days, the 400 mb LCL values were between 393 and 356 mb, extremely moist conditions, though no precipitation was measured at Phortse. The plumes were observed to stream well into Tibet. The plumes probably appeared as those in Figure 9; initially liquid then forming snow particles downwind. This plume outbreak caused the average plume formation time in December 2020 (Table 2) to be 0700 LST.*

*The following paragraph will be added to the Conclusions:*

*The Everest massif was visible on 143 of the 151 observation days (95%), especially in the morning because the plumes most often formed later in the morning. On the days the massif was visible, plumes were observed to form on 63 days (44%). The plumes lasted an average of 12 hours. Of these plumes, 59 (94%) were predicted to form and 4 (6%) were not predicted. These four plumes were not composed of resuspended snow because they were not visible at sunrise. Though, plumes of resuspended snow may have occurred smaller than our detection limit of a couple of kilometers. Snowfall from the plumes, formed insitu, was estimated and may be significant.*

*The following sentence will be inserted in the abstract on line 12 between 'wake.' and 'We':*

*The massif was visible on 143 days (95%), plumes formed on 63 days (44%) and lasted an average of 12 hours.*

Also, the authors mention in the conclusions that the Everest plumes may be a source of snowfall. They didn't mention that before in the manuscript. A rough estimation of how much the plumes contribute to the snow fall would benefit the manuscript.

*To address this insightful comment, we will insert the following section:*

**4.3 Estimate of snowfall from the observed plumes**

*Assume a saturated parcel of air ascends moist adiabatically from the elevation of the South Col of Everest (~7900 m, ~400 mb) to the summit (~8900 m, 300 mb). The parcel is initially -33 $^0$C (the average plume temperature, Table 2) and cools to -40 $^0$C at the summit. The initial parcel saturated mixing ratio is 0.59 g/kg and the final is 0.39 g/kg for an average of 0,49 g/kg. Employing the precipitable water calculator at http://www.shodor.org/os411/courses/_master/tools/calculators/precipwater/, ~1 mm of water is expected to precipitate from the parcel.*

*Assume the parcel ascends in the turbulent wake the 1000 m from the South Col to the summit at 0.1 m/s, the ascent takes $10^4$ s. So, every $10^4$ seconds 1 mm of liquid precipitates from the parcel. The average duration of the observed plumes was 12 hours (Table 2) or 4.32 x $10^4$ seconds. The amount of precipitation from the average plume was 1 mm/$10^4$ s x 4.32 x $10^4$ s or about 4 mm.*

*Sixty-three (63) Everest plumes occurred during our four-month observation period (Table 2). So, 63 plumes x 4 mm/plume equals about 252 mm (~25 cm) of liquid-equivalent may have precipitated. The amount of liquid-equivalent precipitation measured at Phortse during our observation period was 284.5 mm (~28 cm). Thus, Everest plumes may be a significant source of precipitation.*

The authors mentioned the visible differences between resuspended snow and banner clouds (line 51), which were shown in Schween et al. (2007). I was wondering if this might be another indicator to support the analysis, if pictures from Mount Everest would be available. But according to line 65, continuous imaging is not available. I was wondering about that and found a video live stream

(https://www.skylinewebcams.com/de/webcam/nepal/khumbu-pasanglhamu/khumjung/mount-everest.html ),

which shows a view of Mount Everest from a similar position as in Figure 3. I think it would be possible to develop a simple program which takes snapshots of this stream and identify the plumes in addition to satellite images. Maybe that would help for a follow up study.

*Thank you for identifying this live-stream feed from the HEV, the same location as our 1995-11-28 time-lapse movie associated with Figure 3. We contacted the HEV and learned this feed began in January 2022 and is not archived. So, unfortunately, the feed cannot help our study.*

The title is "The Mount Everest plume in winter". If you would change it to "The formation and composition of the Mount Everest plume in Winter" the reader would already have an idea what you are going to analyze.

*Good suggestion, we will change the title as you suggest.*

**Minor comments**:

Line 8: „plume often forms"…. How often?

*"'often forms' will be replaced by 'can form'.*

Line 10: „collect the corresponding meteorological data"… What kind of data?

*Lines 9 through 11 will be revised to read:*

> *Accordingly, daily, we observed real-time images from a geosynchronous meteorological satellite from 1 November 2020 through 31 March 2021 (151 days) to identify the days plumes formed. Daily, surface and upper-air meteorological data were collected.*

Line 18: „is the highest elevation" …. How high?

*In line 18, the following value will be inserted after 'elevation': (8856 m)*

Fig. 2 to Fig. 8: All Figures need panel labels like „a) , b), c)". That would make it much easier for the reader to find the panel.

*Will do.*

Line 88: „400mb" and later in line 90 „300 mb". A space between number and unit is correct. Try to be consistent through the manuscript.

*This comment will be followed.*

Line 99: „-35C" … it is -35 °C. This needs to be changed in the whole manuscript.

*Will do.*

Figure 4 and 5: The profiles are too small and the resolution is too bad. It is not possible to identify the numbers.

*Here is revised Fig. 4; Fig. 5 will be revised similarly:*

[Figure]

*Figure 4: The images and profiles, a) to c), are for 2021-01-25, -26 and -27 at 15LST (Local Solar Time) or 09Z. The locations of the major peaks are circled. The lifting-condensation-level (LCL) values are determined graphically on the corresponding atmospheric profiles from Phortse and are listed in Table 1. The graphical procedures are described in the text. The approximate pressures at the base and summit of the Everest pyramid, respectively, are approximately 400 and 300mb.*

All Satellite images: It would be nice to have a km scale bar on the pictures to see the dimension. Same for the Videos.

*In response to this comment, the following paragraph will be inserted between lines 77 and 79:*

> *To locate the features and know distances in the H-8 images and movies, a map is given in Figure 1a. The map provides a distance scale and identifies the locations of the major mountain peaks, the HEV, Phortse and the Arun Valley. The times and dates for all the H-8 images are displayed on the movies and the images themselves.*

Citation: https://doi.org/10.5194/acp-2021-966-RC2

---

## Author Response (AR1)

**RC1 comments on 'acp-2021-966' by Volkmar Wirth, 31 Jan 2022**

The current paper investigates the occurrence and origin of plumes in the lee of major mountain peaks in the Himalayas. The authors combine satellite imagery from a few episodes with temperature and humidity data at a station close to Mt Everest. They conclude that most of the observed plumes are, indeed, cases of so-called banner clouds, meaning they have been generated by condensation of water vapor in an ascending air stream in the lee of the mountain (rather than just snow being blown off the summit).

The method to analyze the meteorological situation is straightforward: the authors compute the lifting condensation level (LCL) from temperature and moisture data at the observational site, and then check whether the LCL is below or above the summit of the respective mountain. If the LCL is below the summit, they hypothesize that one should expect a banner cloud to occur. Comparing this "expectation" with observations from satellite images indicates good agreement. This then allows the authors to conclude that the plumes observed on the satellite images are, indeed, real banner clouds.

This is the first study of its kind that I am aware of, and I think it is worth a publication in a science journal. My only criticism is that the spatial resolution of the satellite imagery is marginally low for the intended purpose: for me as a reader it required occasionally quite a bit of imagination to see what the authors see. Yet, the movies improve upon this issue, as the non-stationarity of (even a poorly resolved) feature in these movies allows one to more or less clearly identify the presence of a plume.

Below I have two major issues as well as a number of minor issues. My remarks are meant to help to produce a revised version.

Volkmar Wirth

*We thank Prof. Wirth for his careful and thorough reading of our manuscript. We respond to his insightful comments in italics:*

**Major issues:**

I think it would be best to be explicit and honest about the low resolution of the satellite imagery. I.e., you best are honest and admit that the resolution is marginal, but (in particular in combination with the animations) just about sufficient to draw the conclusions that you want to draw.

*To respond to this issue, the following paragraphs will be inserted between lines 71 and 73:*

> *The spatial resolution of the H-8 images is sufficient to resolve the plumes, not as they form, but shortly thereafter. The following is our reasoning. The sub-satellite point is at 0N, 104.7E and the summit of Mount Everest is at 27.99N, 86.93E. At the sub-satellite point, the satellite zenith angle is 0 degrees (nadir) and the spatial resolution is 0.5 km for images in the visible Band 3 and 2.0 km for images in the infrared Band 13. Careful examination of pixel edges suggests that the 0.5 km and 2 km nadir resolutions are degraded to, respectively, about 1 km and 4 km in the vicinity of Everest. The plume Moore (2004) studied, shown in Figure 2, he estimated to be 15 km in length. Also comparing the plumes in Figure 3 with the map in Figure 1a [figure is shown in our replies to RC2 comments], it can be seen that the plumes were kilometers in length. So, had the H-8 been in orbit in 1992 and 2004, these plumes would have been observed.*

> *The images from the H-8 website were observed daily in the both the 'still' and 'animation' modes. The images could be magnified 300X on the FireFox browser and the site provided animations up to 23-hours. The forming plumes were observed as moving elements against a mostly stationary background. Once*

*they reached a couple of kilometers in length, the lengthening and undulations of the plumes, shown in Movie 1, were observed.*

*To permit the reader to observe the formation and development of the plumes, we present movies made from the every-ten-minute H-8 images. All of the H-8 images presented here are oriented such that the vertical points toward true north; Fig. 1c is a map of the region [figure is shown in our replies to RC2 comments]. The map provides a distance scale and identifies the locations of the major mountain peaks, the HEV, Phortse and the Arun Valley. The times and dates for all the H-8 images are displayed on the images and the movies themselves. The images and movies were produced following procedures in the Data Availability section.*

In your introduction you state that your results are "conclusive" (as opposed to earlier results); in my view this is somewhat overstated. To be sure, you add some circumstantial evidence, but this is not extremely convincing to me owing to the low resolution of the satellite imagery.

*In line 44, "concluded" will be replaced with "reported".*

Also, I suggest that you systematically distinguish between the concept of a plume and the concept of a banner cloud. For me, a plume is anything that you see in the lee of a steep and high mountain (including snow blown off the mountain top). By contrast, a banner cloud is a plume that has been generated through condensation of moisture in an upwelling airstream in the immediate lee of the mountain.

*As a result of this comment, lines 48 through 52 will be deleted. And, the following paragraph will replace lines 60 through 63.*

*Schween and colleagues (2007) show still images and animations, all with the same view, from the summit of the Zugspitze in the Bavarian Alps. Because of the best possible spatial and temporal resolution, they were able to show the formation of banner clouds and snow blown off an adjacent peak. Here we use the best possible spatial and temporal resolution images available to us from meteorological satellites to observe the formation of plumes in the lee of the Everest massif. The plumes most likely were banner clouds when our calculations predicted cloud formation through condensation of moisture in the airstream upwelling in the immediate lee of the massif.*

**Minor issues:**

The movies: It would be good if you could provide movie-captions for all the movies.

*The movies of the Himawari-8 images (Movies 2, 3 and 4) already are sufficiently captioned with time, date and image band. Significant times in Movie 1 were inserted as captions.*

Movie 1: it would be good to know how local time evolves as the movie passes by; is it possibly to include a little clock running with the movie accordingly? Or to give at least the time span (beginning and end time) covered by the movie.

*It was not possible to insert a 'little clock'. And the movie was captioned following this paragraph that will be inserted between lines 47 and 48:*

*Movie 1 captures the formation and evolution of the plume: The movie began at 0940LST showing the summits of Everest (poking over the Nuptse ridge) and Lhotse (to the right) were plume free. At about 1050LST, a plume began in the wake of Lhotse. Clouds began to form on the valley slopes about 1200LST. The plume reached full development at about 1400LST. At that time, the plume began to be*

*intermittently obscured by clouds filling the valley. The movie ends at 1630LST because the HEV was enveloped by the clouds that had completely filled the valley.*

The satellite movies are very coarse resolution. Is there a possibility to post-process them in order to more clearly focus on what you want to show? (Maybe not, indeed, because nothing beats the pattern recognition skills of the human brain.)

*In response to this excellent suggestion, we will add the following paragraph between lines 311 and 313 in the 'Data availability' section of our manuscript:*

> *Wirth (2022) suggested we attempt to post-process the best-resolution H-8 visible imagery to improve the movie resolution. In general, the sharpening techniques we are aware of (in SatPy for example) require a higher resolution band. So, for example on H-8, Band 1 (0.47 micrometers, with 1-km resolution at nadir) or Band 2 (0.51 micrometers, also 1-km resolution) can be sharpened with information from Band 3 (0.64 micrometer, with 0.5-km resolution at nadir). So, there is no practical method to improve the spatial resolution in Band 3.*

*Wirth (2022) will be added to the reference list as "reviewer comment, doi.org/10.5194/acp-2021-966-RC1".*

Table 1: I think it would be better to provide wind speed in m/s rather than knots.

*The speeds in m/s will be added to Table 1.*

Line 48: Sentence unclear to me.

*As a result of this comment, lines 48 through 52 will be deleted. And, the following paragraph will replace lines 60 through 63.*

> *Schween and colleagues (2007) show still images and animations, all with the same view, from the summit of the Zugspitze in the Bavarian Alps. Because of the best possible spatial and temporal resolution, they were able to show the formation of banner clouds and snow blown off an adjacent peak. Here we use the best possible spatial and temporal resolution images available to us from meteorological satellites to observe the formation of plumes in the lee of the Everest massif. The plumes most likely were banner clouds when our calculations predicted cloud formation through condensation of moisture in the airstream upwelling in the immediate lee of the massif.*

Line 62: See our paper Prestel and Wirth (2016) where we elucidate the conditions under which one would expect a banner cloud to occur (steep mountain, week stratification).

*In line 54, Prestel and Wirth (2016) will be inserted after Voigt and Wirth (2013).*

Line 68: what resolution are these satellite images? Is it good enough to well resolve the cloud?

*This issue was addressed with the paragraphs previously inserted between lines 71 and 73.*

Line 79: which model? More details!

*Our response to this issue will be to replace Lines 79 through 83 with the following:*

> *It can be seen in Fig. 1a [the figure is shown in a RC2 comment], that both Everest and its neighbor to the south, Lhotse, present significant obstacles to the typically west-to-east air flow. Hence, both peaks*

*produce wakes and, as seen in Fig. 2-top, both produced plumes.  Cloud formation was investigated in the dynamically-forced lee upslope flow in these wakes.  The lifted-condensation-level (LCL) of the upslope flow was calculated with the following procedure.*

Line 82: "Hence….": do you want to imply that banner clouds occur only on pyramid-shaped mountains?

*Line 82 will be replaced by the statement replacing Lines 79 through 83 that does not include the words 'summit pyramid'.*

Line 96: What do you mean by "initial composition" here? (It becomes clear somewhat later….).

*Our response to this issue will be to replace Line 96 with the following:*

> **The composition of a forming plume was inferred from the temperature at the LCL.**

Can you exclude the possibility that these clouds are mixed-phase clouds?

*Our response to this insightful statement will be to insert the following paragraph after line 100:*

> *A mixed-phase plume (coexisting droplets and crystals) could not be determined because, at present, the observer must be immersed in the plume.  When Everest experiences a westerly wind, climbers of the SE Ridge, the East Face (Kangshung Face) and the NE ridge could make the observation because the wake forms between these ridges and covers the Kangshung Face (Figure 1b).  During his climb of the Kangshung Face, Venables (1989) recorded numerous meteorological observations. But, he did not report being immersed in a fog and seeing scintillations from forming, pristine ice crystals.*

Fig. 4: Make the Tephigrams larger, they are important!

*Here is revised Fig. 4 (it's displayed full-scale in the accompanying 'clean copy'):*

[Figure]

*Figure 4: The images and profiles, a) to c), are for 2021-01-25, -26 and -27 at 15LST (Local Solar Time) or 09Z.  The locations of the major peaks are circled.  The lifting-condensation-level (LCL) values are determined graphically on the corresponding atmospheric profiles from Phortse and are listed in Table 1.  The graphical procedures are described in the text.  The approximate pressures at the base and summit of the Everest pyramid, respectively, are approximately 400 and 300mb.*

Fig. 6: There seems to be a problem/mismatch between the yellow caption inside the image and the added caption below the image in the bottom row left and middle column.

*Here is the corrected Figure 6 (it's displayed full-scale in the accompanying 'clean copy'):*

[Figure]

Line 146 ff: I found it hard to verify the description/interpretation that you provide in the text when viewing the images.

*The following words will be inserted in line 147 to make the sentence read:*

*As observed in Event 1, sharp shadows cast……,*

Line 164: 4C? Do you mean 4 degrees Celsius?

*Yes*

Line 182/183: Haven't you said something very similar a few lines earlier?

*Woops! Lines 182 and 183 will be deleted.*

Line 204: What do you mean if you mention a "Jet stream…. embedded in a trough…."?

*The importance of Figure 7 will be clarified by modifying the sentence in lines 204 and 205 to read:*

*These winds were caused by the jet-stream that moved through the Everest region during the 8th and 9th as shown by the sequence of images in Fig. 7. The red sinuous region defines the jet stream. Additionally, it can be seen in the sequence the trough of the Western Disturbance, in which the jet stream was embedded, was east of the Everest region and had moved slowly eastward.*

Figure 7: What do you want to clarify by showing this sequence of maps? Is the evolution important? Could you show just one panel as representative for the entire episode?

*The evolution is important as described above in your Line 204 question.*

Figure 9: Apparently, upward is not northward in these satellite images. Please notify the reader in the figure caption accordingly.

*The following will be inserted in the caption:*

*Figure 1c [shown in 'RC2 comments' below] is a map of the region displayed in this image.*

Also, it would be nice if you could somehow indicate the northward direction on these satellite images.

*To resolve this issue, the following sentence will be added to the last paragraph of the paragraphs inserted at line 71:*

*All of the H-8 images presented here are oriented such that the vertical points toward true north as seen in Fig. 1c.*

**RC2 comments on 'acp-2021-966', Anonymous Referee #2, 10 Feb 2022**

The manuscript "The Mount Everest plume in Winter" by E. E. Hindman and S. Lindstrom analyses the formation and composition of the Mount Everest plume. To do so, the authors examined two wintertime plumes in detail by using GMS images and atmospheric sounding data, which were used for an atmospheric model simulation. The results of this work conclude that the identified plumes are either composed of cloud droplets or ice particles, but were not composed of resuspended snow.

The authors describe their analysis method very clearly and summarize the history of plume studies at Mount Everest in a very nice way. Their analysis method is simple, but was never done before for the Mount Everest plume. For this reason, I recommend to publish this work in ACP after a major revision.

*We thank this reviewer for their careful and thorough reading of our manuscript. We respond to their comments in italics:*

**Major comments**:

This study is mainly based on satellite images with a very low resolution. For me as a reader, it was very hard to see the described observations on the images and especially the videos. I don't doubt their method, but I think it is worth it to put some more effort in the presentation of the images and videos to make the authors' statements clearer. I would recommend to add labels with time, km scale bars and markers for the location of the summits.

*Our response to this comment is included in the response to RC1 in the text inserted between lines 71 and 73:*

*Also, Figure 1 will be expanded to illustrate the important features of the Mt. Everest region presented in the manuscript (full-scale image is in the accompanying 'clean' manuscript:*

[Figure]

*Figure 1. (a) The Mount Everest and Lhotse summit pyramids are outlined. The bases of the pyramids are at an elevation of approximately 7900 m. The summits are, respectively, 8848 m and 8501 m in elevation. The map is from the November 1988 issue of the National Geographic Magazine. (b) The Everest summit pyramid at*

*sunrise in May 2010 as viewed from near the summit of Lhotse (from CoryRichards.com and Anker, et. al. (2013)). (c) The Mount Everest region with the major summits and locations identified (the HEV is the Hotel Everest View; the chart is from skyvector.com).*

*The Anker, et. al. reference will be added to the list: Conrad, A., et. al., The call of Everest. National Geographic Society, 303 pp. ISBN 978-1-4262-1016-7, 2013.*

The authors are focusing on the development process of the plumes and describe that the observed plumes and the Moore plume are not composed of resuspended snow. It gives the impression that plumes from resuspended snow are not possible at all, which I don't think the authors intend. I think the Authors should make clear that this is just a case study over a few days (see Table 1) and plumes of resuspended snow might be still possible.

*In response to this insightful comment, the following text will be inserted between line 111 and 113:*

> *We recorded the days the Everest massif was observed to produce a plume, the formation time of the plume, the plume duration and how many plume events were predicted by the LCL model. Cases where a plume was observed but not predicted are investigated because they might be plumes of resuspended snow.*

I would also be interested in some more statistics. For example, how often do these plumes occur per month and how long do they usually persist?

*To answer this important suggestion, a new section '3.4 Plume statistics' will be inserted before Section 3.4:*

**3.4 Plume statistics**

Table 2

Results from the observations of Himaeari-8 imagery and the Lifted Condensation Level (LCL) calculations from the corresponding atmospheric profiles (yellow shading denotes summed values, green shading denotes average values)

| Month 2020-21 | Number of days observed | Everest visible | Plume observed | Average plume formation time (hour LST) | Average plume duration (hours) | Average LCL temperature ($^0$C) | Aver age 300 mb winds (degrees) | (m/s) | Plume predicted | Plume not predicted |
|---|---|---|---|---|---|---|---|---|---|---|
| November | 30 | 26 | 7 | 10 | 8 | -32 | 271 | 38 | 7 | 0 |
| December | 31 | 31 | 15 | 7 | 14 | -31 | 268 | 47 | 14 | 1 |
| January | 31 | 31 | 7 | 9 | 14 | -31 | 266 | 45 | 6 | 1 |
| February | 28 | 28 | 17 | 9 | 11 | -35 | 247 | 28 | 15 | 2 |
| March | 31 | 27 | 17 | 9 | 11 | -34 | 269 | 32 | 17 | 0 |
| Sum | 151 | 143 | 63 | 9 | 12 | -33 | 264 | 38 | 59 | 4 |
|  |  | 95% | 44% |  |  |  |  |  | 94% | 6% |

*Table 2 displays the results from our 151 daily observations of H-8 imagery and the corresponding 400 mb LCL values calculated from the atmospheric profiles. It can be seen from the table, Everest was almost always visible (95%), especially in the morning because the plumes most often formed later in the morning. On almost half of the days Everest was visible (143 days), plumes were observed to form on 63 days (44%). Of these plumes, 59 (94%) were predicted to form and 4 (6%) not predicted. Were the four plumes composed of resuspended snow?*

*Here are the steps we used to determine if a plume was a banner cloud or resuspended snow. Scrolling frame-by frame through the Movie 4 (Event 3) shows the plume was not visible as the sun rose and was clearly visible as the sun set. This behavior was consistent with the LCL results: the LCL was above the Everest summit in the morning and below the summit in the afternoon; this plume was a banner cloud. If this plume had been visible at sunrise and the corresponding LCL value was above the Everest summit, then the plume was composed of resuspended snow. These steps are applied to the four plumes suspected to be blowing snow.*

*The four plumes were observed on 2020-12-05, 2021-01-29 and 2021-02-03 and 11. The 400 mb LCL values ranged between 295 to 249 mb, all above the 300 mb level of the Everest summit. The plumes formed between*

1200 and 1400 LST and dissipated around 1900 LST. The plumes were not visible at sunrise and visible at sunset. Therefore, these plumes were not composed of resuspended snow. Thus, none of the 63 plumes we observed we conclude were composed of resuspended snow. Though, plumes of resuspended snow may have occurred smaller than our detection limit of a couple of kilometers.

[To explain the following paragraph, please see our comments below dated 2022-04-29]

Twice-daily images of the Everest summit coincident with a portion of our H-8 observations became available from Grey, et al. (2022) while this study was in peer-review. The images were taken from 2020-12-16 through 2021-01-16 (32 days) at ~10 and ~17 LST. We studied the images to determine the number of days the summit was visible and the number of days plumes occurred. The summit was visible on 28 days (88%) while the corresponding H-8 observations revealed the massif was visible on 32 days (100%). The summit produced 18 morning plumes and 11 afternoon plumes. The corresponding H-8 observations detected 8 of the morning plumes and 4 of the afternoon plumes. This comparison shows a number of Everest plumes did not reach the couple-of-kilometers in length to be detected in the real-time H-8 images.

We observed plumes we suspect were composed primarily of snow formed in situ as shown in Movie 5. The movie was constructed from the real-time H-8 infrared images as described in the Data Availability section. Note, in the movie NT is Nepal Time which is approximately LST. The major summits are labelled and are seen as white, stationary objects. On 2020-12-21 plumes formed in the morning downwind of the Everest massif and Cho Oyu. The plumes are seen to dissipate four days later on 2020-12-25 early in the morning. The plumes fluctuated in length and can be seen to stream well into Tibet. The 400 mb LCL values were between 393 and 356 mb, extremely moist conditions, though no precipitation was measured at Phortse.

The following paragraph will be added to the Conclusions by replacing the paragraph beginning on Line 293:

> The Everest massif was visible on 143 of the 151 observation days (95%), especially in the morning because the plumes most often formed later in the morning. On the days the massif was visible, plumes were observed to form on 63 days (44%). The plumes lasted an average of 12 hours. Of these plumes, 59 (94%) were predicted to form and 4 (6%) were not predicted. These four plumes were not composed of resuspended snow because they were not visible at sunrise. Though, plumes of resuspended snow may have occurred smaller than our detection limit of a couple of kilometers. Snowfall from the plumes, formed insitu, was estimated and may be significant.

The following sentence will be inserted in the abstract in Line 12 between 'wake.' and 'We':

> The massif was visible on 143 days (95%), plumes formed on 63 days (44%) and lasted an average of 12 hours.

Also, the authors mention in the conclusions that the Everest plumes may be a source of snowfall. They didn't mention that before in the manuscript. A rough estimation of how much the plumes contribute to the snow fall would benefit the manuscript.

To address this insightful comment, we will insert the following section after Line 283:

**4.3 Estimate of snowfall from the observed plumes**

Assume a saturated parcel of air ascends moist adiabatically from the elevation of the South Col of Everest (~7900 m, ~400 mb) to the summit (~8900 m, 300 mb). The parcel is initially -33 $^0C$ (the average plume temperature, Table 2) and cools to -40 $^0C$ at the summit. The initial parcel saturated mixing ratio is 0.59 g/kg and the final is 0.39 g/kg for an average of 0,49 g/kg. Employing the precipitable water calculator at

*http://www.shodor.org/os411/courses/_master/tools/calculators/precipwater/, ~1 mm of water is expected to precipitate from the parcel.*

*Assume the parcel ascends in the turbulent wake the 1000 m from the South Col to the summit at 0.1 m/s, the ascent takes $10^4$ s. So, every $10^4$ seconds 1 mm of liquid precipitates from the parcel. The average duration of the observed plumes was 12 hours (Table 2) or 4.32 x $10^4$ seconds. The amount of precipitation from the average plume was 1 mm/$10^4$ s x 4.32 x $10^4$ s or about 4 mm.*

*Sixty-three (63) Everest plumes occurred during our four-month observation period (Table 2). So, 63 plumes x 4 mm/plume equals about 252 mm (~25 cm) of liquid-equivalent may have precipitated. The amount of liquid-equivalent precipitation measured at Phortse during our observation period was 284.5 mm (~28 cm). Thus, Everest plumes may be a significant source of precipitation.*

*The plume-generated snowfall is expected to be a maximum in the immediate lee of the Everest massif and diminish downwind as drier air is entrained. The always-white Kangshung face of Everest (Fig. 1b) may be evidence of plume-generated snowfall, although, much of this snow probably may be captured from snow-filled clouds flowing around the summit pyramid. This capture is similar to snow collecting on the tailgate of a truck speeding through a snow storm.*

The authors mentioned the visible differences between resuspended snow and banner clouds (line 51), which were shown in Schween et al. (2007). I was wondering if this might be another indicator to support the analysis, if pictures from Mount Everest would be available. But according to line 65, continuous imaging is not available. I was wondering about that and found a video live stream

(https://www.skylinewebcams.com/de/webcam/nepal/khumbu-pasanglhamu/khumjung/mount-everest.html ),

which shows a view of Mount Everest from a similar position as in Figure 3. I think it would be possible to develop a simple program which takes snapshots of this stream and identify the plumes in addition to satellite images. Maybe that would help for a follow up study.

*Thank you for identifying this live-stream feed from the HEV, the same location as our 1995-11-28 time-lapse movie (Movie 1). We contacted the HEV and learned this feed began in January 2022 and is not archived. So, unfortunately, the feed cannot help our study. Consequently, we replace the sentence in Line 66 with the following sentence:*

> *[Note: Anonymous reviewer (2022) informed us of a live-stream of the massif from the HEV (www.youtube.com/watch?v=RgDjOg4WvGI). The stream was not useful for this study because it began in January 2022)].*

The title is "The Mount Everest plume in winter". If you would change it to "The formation and composition of the Mount Everest plume in Winter" the reader would already have an idea what you are going to analyze.

*Good suggestion, we will change the title as you suggest.*

**Minor comments**:

Line 8: „plume often forms"…. How often?

> *"'often forms' will be replaced by 'can form'.*

Line 10: „collect the corresponding meteorological data"… What kind of data?

*Lines 9 through 11 will be revised to read:*

> *Accordingly, daily, we observed real-time images from a geosynchronous meteorological satellite from 1 November 2020 through 31 March 2021 (151 days) to identify the days plumes formed. Daily, surface and upper-air meteorological data were collected.*

Line 18: „is the highest elevation" …. How high?

*In line 18, the following value will be inserted after 'elevation': (8856 m)*

Fig. 2 to Fig. 8: All Figures need panel labels like „a) , b), c)". That would make it much easier for the reader to find the panel.

*Will do.*

Line 88: „400mb" and later in line 90 „300 mb". A space between number and unit is correct. Try to be consistent through the manuscript.

*This comment will be followed.*

Line 99: „-35C" … it is -35 °C. This needs to be changed in the whole manuscript.

*Will do.*

Figure 4 and 5: The profiles are too small and the resolution is too bad. It is not possible to identify the numbers.

*Fig. 4 and Fig. 5 will be revised as shown in our reply to a comment by RC1.*

All Satellite images: It would be nice to have a km scale bar on the pictures to see the dimension. Same for the Videos.

*This comment was addressed in the final paragraph inserted between lines 71 and 73 in response to a comment by RC1.*

Line 228: Am I supposed to see that shadow in the satellite image?

*Yes. The shadow is labeled in revised Figure 8 (full-scale of this image id in the accompanying 'clean' manuscript):*

[Figure]

*Figure 8. GOES-9 0.65 micrometer images a) through f) for local solar time (LST) on 28 and 28 January 2004. The major features are labelled.*

Line 232/233: For me this is hard to see on the satellite image.

*The following sentence will be added to the end of line 233:*

> *The protruding plumes are difficult to identify in Figure 8. So, we searched the archives for finer spatial-resolution images from polar orbiting satellites.*

Line 244: The difference between sharp and fuzzy is hard for me to see on the satellite image.

*The following sentence will be inserted after the sentence ending in 'suggesting glaciation':*

> *The regions of the plumes containing primarily cloud droplets are the most reflective hence the brightest, the whitest. The region of the plume containing primarily much larger ice crystals are less reflective and appear dimmer and grayer.*

Line 259: here you could specifically point to their Fig. 5b, which explicitly shows the diurnal cycle at Mt Zugspitze.

*In line 259, 'Wirth, et. al. (2012)' will be expanded to read 'Wirth, et. al. (2012, Fig. 5b)'.*

Line 290: "…. Presented evidence…", well, rather weak evidence, essentially based on the interpretation of a very low-resolution satellite image.

*The statement 'very low-resolution' is not correct. The statement also was made by RC1 and addressed with the text inserted in Line 236.*

Line 295: "… expect the plumes to form …." not clear whether I understand the logic behind this argument.

*The paragraph in which this statement appears will be replaced by the following paragraph:*

> *The Everest massif was visible on 143 of the 151 observation days (95%), especially in the morning because the plumes most often formed later in the morning. On the days the massif was visible, plumes were observed to form on 63 days (44%). The plumes lasted an average of 12 hours. Of these plumes, 59 (94%) were predicted to form and 4 (6%) were not predicted. These four plumes were not composed of resuspended snow because they were not visible at sunrise. Though, plumes of resuspended snow may have occurred smaller than our detection limit of a couple of kilometers.'*

*The paragraph beginning on line 299 was replaced by the following sentence:*

> *The plume studied by Moore (2004) we show was a banner cloud, not a plume of resuspended snow.*

**2022-04-29**

Dear Peer reviewer:

On 2022-03-16, Scott Lindstrom and I submitted our revised manuscript **acp-2021-966** *The formation and composition of the Mount Everest plume in winter*. The manuscript addressed the two peer reviewers' comments.

On 2022-03-22, while our manuscript was in peer-review, an article was published in *Weather* that provided systematic images of the Everest summit from the surface coincident with a portion of the real-time images we observed from the Himawari-8 (H-8) geosynchronous satellite. We analyzed the images in *Weather* to determine the number of days the summit was visible and the number of days plumes occurred. We found the H-8 images did not resolve a number of the plumes detected in the images from the surface. We concluded the plumes not detected did not reach the required couple-of-kilometers in length.

These new findings were inserted in the Line 211 comment in the marked-up copy between the last two paragraphs:

> Twice-daily images of the Everest summit coincident with a portion of our H-8 observations became available from Grey, et al. (2022) while this study was in peer-review. The images were taken from 2020-12-16 through 2021-01-16 (32 days) at ~10 and ~17 LST. We studied the images to determine the number of days the summit was visible and the number of days plumes occurred. The summit was visible on 28 days (88%) while the corresponding H-8 observations revealed the massif was visible on 32 days (100%). The summit produced 18 morning plumes and 11 afternoon plumes. The corresponding H-8 observations detected 8 of the morning plumes and 4 of the afternoon plumes. This comparison shows a number of Everest plumes did not reach the couple of kilometers in length to be detected in the real-time H-8 images.

The following paragraph was inserted in the Conclusions at Line 297:

> Our analysis of the Grey et. al. (2022) images of the Everest summit from the surface showed a number of Everest plumes did not reach the couple of kilometers in length to be detected in the real-time H-8 images. Thus, our plume-occurrence values should be considered a lower-limit.

The *Weather* reference was inserted in the Reference list before the Hindman and Wick reference.

On 2022-03-16 we submitted Movie 5 that illustrates extremely long plumes produced downwind of Cho Oyu and the Everest massif. Thereafter, we produced a copy with regular time captions:

> improved Movie 5.

The improved video has been submitted to Copernicus. Please view that movie.

Thank you for reviewing our manuscript, a sacred collegial duty!

Sincerely,

Edward Hindman